# Stealthy Shield Defense: A Conditional Mutual Information-Based Approach Against Black-Box Model Inversion Attacks

**Tianqu Zhuang[1*], Hongyao Yu[2*], Yixiang Qiu[1*], Hao Fang[1*], Bin Chen[2#], Shu-Tao Xia[1]**

[1]Shenzhen International Graduate School, Tsinghua University, China
[2]Harbin Institute of Technology, Shenzhen, China
{zhuangtq23, qiu-yx24, fang-h23}@mails.tsinghua.edu.cn; yuhongyao@stu.hit.edu.cn;
chenbin2021@hit.edu.cn; xiast@sz.tsinghua.edu.cn; *Equal contribution #Corresponding author

## Abstract

Model inversion attacks (MIAs) aim to reconstruct the private training data by accessing the public model, raising concerns about privacy leakage. Black-box MIAs, where attackers can only query the model and obtain outputs, are closer to real-world scenarios. The latest black-box attacks have outperformed state-of-the-art white-box attacks, and existing defenses cannot resist them effectively. To fill this gap, we propose Stealthy Shield Defense (SSD), a post-processing algorithm against black-box MIAs. Our idea is to modify the model's outputs to minimize the conditional mutual information (CMI). We mathematically prove that CMI is a special case of Information Bottleneck (IB), and thus inherits the benefits of IB—making predictions less dependent on inputs and more dependent on ground truths. This theoretically guarantees our effectiveness, both in resisting MIAs and preserving utility. To minimize CMI, we formulate a convex optimization problem and solve it via the water-filling method. Without the need to retrain the model, our defense is plug-and-play and easy to deploy. Experimental results indicate that SSD outperforms existing defenses, in terms of MIA resistance and model's utility, across various attack algorithms, private datasets, and model architectures. Our code is available at `https://github.com/ZhuangQu/Stealthy-Shield-Defense`.

## 1 Introduction

Deep neural networks (DNNs) have driven widespread deployment in multiple mission-critical domains, such as computer vision (He et al., 2015), natural language processing (Devlin et al., 2019) and dataset distillation (Zhong et al., 2024b;a). However, their integration with sensitive training data has raised concerns about privacy breaches. Recent studies (Fang et al., 2024b;a; 2025) have explored various attack methods to probe such privacy, such as gradient inversion (Fang et al., 2023; Yu et al., 2024b) and membership inference (Hu et al., 2021). Among the emergent threats, model inversion attacks (MIAs) aim to reconstruct the private training data by accessing the public model, posing the greatest risk (Qiu et al., 2024b). For instance, consider a face recognition access control system with a publicly accessible interface. Through carefully crafted malicious queries, model inversion attackers can infer the sensitive facial images stored in the system, along with the associated user identities.

MIAs are divided into *white-box* and *black-box* (Fang et al., 2024c). White-box attackers know the details of the model, whereas black-box attackers can only query the model and obtain outputs. Black-box MIAs become more threatening than white-box because: **(1) Black-box scenarios are more common.** As models grow larger nowadays, they are mostly stored on servers and can only be accessed online, which is a typical black-box scenario. **(2) Black-box attacks are more powerful.** The latest soft-label attack RLBMI (Han et al., 2023) and hard-label attack LOKT (Nguyen et al., 2023) have outperformed state-of-the-art white-box attacks. **(3) Existing defenses cannot resist black-box attacks effectively.** Existing defenses focus on modifying the weights and structure of the model, but black-box attackers only exploit the outputs, and thus are less susceptible.

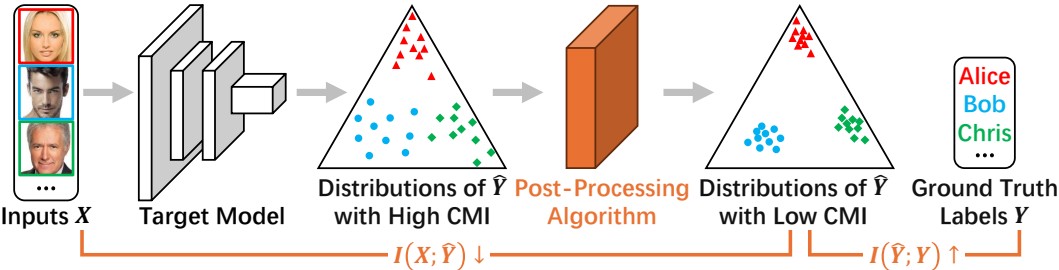

Figure 1: An overview of Stealthy Shield Defense. With 3 classes, the probability simplex is a triangle. CMI is defined as $\mathcal{I}(X; \hat{Y}|Y)$. According to our Theorem 1, minimizing CMI makes the mutual information $\mathcal{I}(X; \hat{Y})$ minimized and $\mathcal{I}(\hat{Y}; Y)$ maximized. As shown by Yang et al. (2024), minimizing CMI makes the outputs more concentrated class-wisely.

To address these concerns, we propose Stealthy Shield Defense (SSD), a post-processing algorithm against black-box MIAs. As shown in Figure 1, the idea of SSD is to modify the model's outputs to minimize the conditional mutual information (CMI). CMI quantifies the dependence between inputs and predictions when ground truths are given. In Theorem 1, we prove that CMI is a special case of Information Bottleneck (IB), and thus inherits the benefits of IB—making predictions less dependent on inputs and more dependent on ground truths. Under this theoretical guarantee, SSD achieves a better trade-off between MIA resistance and model's utility. Without the need to retrain the model, SSD is plug-and-play and easy to deploy.

The contributions of this paper are:

- We introduce CMI into model inversion defense for the first time, and theoretically prove its effectiveness.

- We propose a post-processing algorithm to minimize CMI without retraining models. In our algorithm, temperature is introduced to calibrate the sampling probability and adaptive rate-distortion is introduced to constrain the modification to outputs. We speed up our algorithm by GPU-based water-filling method as well.

- Our experiments indicate that we outperform all competitors, in terms of MIA resistance and model's utility, exhibiting strong generalizability across various attack algorithms, private datasets, and model architectures.

## 2 RELATED WORKS

### 2.1 MODEL INVERSION ATTACKS AND DEFENSES

Model inversion attacks (MIAs) are a serious privacy threat to released models (Fang et al., 2024c). MIAs are categorized as *white-box* (Zhang et al., 2019; Chen et al., 2020; Struppek et al., 2022; Yuan et al., 2023; Qiu et al., 2024a) and *black-box*. We focus on black-box MIAs, where attackers can only query the model and obtain outputs. In this scenario, BREP (Kahla et al., 2022) utilizes zero-order optimization to drive the latent vectors away from the decision boundary. Mirror (An et al., 2022) and C2FMI (Ye et al., 2023) explore genetic algorithms. LOKT (Nguyen et al., 2023) trains multiple surrogate models and applies white-box attacks to them.

To address the threat of MIAs, a variety of defenses have been proposed. MID (Wang et al., 2020), BiDO (Peng et al., 2022), and LS (Struppek et al., 2023) change the training losses, TL (Ho et al., 2024) freezes some layers of the model, and CA-FaCe (Yu et al., 2024a) changes the structure of the model. However, black-box attackers only exploit the outputs, and thus are rarely hindered. The defense against black-box MIAs is still limited.

In this paper, we propose a novel black-box defense based on post-processing, without retraining the model. Experimental results indicate that we outperform these existing defenses.

## 2.2 INFORMATION BOTTLENECK AND CONDITIONAL MUTUAL INFORMATION

Tishby et al. (2000) proposed the Information Bottleneck (IB) principle: a good machine learning model should compress the redundant information in inputs while preserving the useful information for tasks. They later highlighted that information is compressed layer-by-layer in DNNs (Tishby & Zaslavsky, 2015; Shwartz-Ziv & Tishby, 2017). Alemi et al. (2017) proposed Variational Information Bottleneck (VIB) to estimate the bounds of IB, and Wang et al. (2020) applied VIB to their Mutual Information-based Defense (MID).

Yang et al. (2024) proposed to use conditional mutual information (CMI) as a performance metric for DNNs, providing the calculation formula and geometric interpretation of CMI. By minimizing CMI, they improve classifiers (Yang et al., 2025) and address class imbalance (Hamidi et al., 2024). By maximizing CMI, they improve knowledge distillation (Ye et al., 2024) and address nasty teachers (Yang & Ye, 2024).

In this paper, we theoretically prove that CMI is a special case of IB and thus inherits the benefits of IB. Furthermore, we propose a novel model inversion defense based on CMI.

## 3 PRELIMINARY

### 3.1 NOTATIONS

Let $f\colon \mathbb{X} \to \mathbb{Y}$ be a neural classifier, $X \in \mathbb{X}$ be an input, $Y \in \mathbb{Y}$ be the ground truth label, $\hat{Y} \in \mathbb{Y}$ be the label predicted by $f$, and $Z \in \mathbb{Z}$ be the intermediate representation in $f$. Note that $Y \to X \to Z \to \hat{Y}$ is a Markov chain. Let $\mathcal{P}$ be the probability function and $\mathcal{P}(x) \coloneqq \mathcal{P}\{X = x\}$, $\mathcal{P}(y) \coloneqq \mathcal{P}\{Y = y\}$, $\mathcal{P}(x, \hat{y}|y) \coloneqq \mathcal{P}\{X = x, \hat{Y} = \hat{y} \mid Y = y\}$, etc.

Let $\Delta^{\mathbb{Y}}$ be the probability simplex over $\mathbb{Y}$, $\boldsymbol{f}(x) \in \Delta^{\mathbb{Y}}$ be the output from the softmax layer when $x$ is input to $f$, and $f_{\hat{y}}(x) \in (0, 1)$ be the $\hat{y}$-th component of $\boldsymbol{f}(x)$, $\hat{y} \in \mathbb{Y}$.

### 3.2 MODEL INVERSION ATTACKS

Let $D \subseteq \mathbb{X} \times \mathbb{Y}$ be the dataset learned by $f$. MIAs aim to reconstruct $\hat{D}$ as close to $D$ as possible. Based on the access to $f$, MIAs are categorized as:

**Hard-label:** Attackers can query any $x \in \mathbb{X}$ and obtain $f(x) \in \mathbb{Y}$, i.e. $\operatorname{argmax}_{\hat{y} \in \mathbb{Y}} f_{\hat{y}}(x)$.
**Soft-label:** Attackers can query any $x \in \mathbb{X}$ and obtain $\boldsymbol{f}(x) \in \Delta^{\mathbb{Y}}$.
**White-box:** Attackers know the details of $f$.

Hard-label and soft-label, collectively called *black-box*,[1] are defended against in this paper.

### 3.3 MUTUAL INFORMATION-BASED DEFENSE (MID)

Wang et al. (2020) proposed to resist MIAs by reducing the dependence between $X$ and $\hat{Y}$. The dependence is quantified by the mutual information, which is defined as

$$\mathcal{I}(X; \hat{Y}) \coloneqq \sum_{x \in \mathbb{X}} \sum_{\hat{y} \in \mathbb{Y}} \mathcal{P}(x, \hat{y}) \log \frac{\mathcal{P}(x, \hat{y})}{\mathcal{P}(x)\mathcal{P}(\hat{y})}. \tag{1}$$

They reduced $\mathcal{I}(X; \hat{Y})$ to prevent attackers from inferring the information about $D$. However, low $\mathcal{I}(X; \hat{Y})$ hurts the model's utility. Especially, $\mathcal{I}(X; \hat{Y}) = 0$ iff $X$ and $\hat{Y}$ are independent, in which case $f$ is immune to any attack but useless at all.

As an alternative, they introduced the Information Bottleneck (Tishby & Zaslavsky, 2015), which is defined as

$$\mathcal{I}(X; Z) - \beta \cdot \mathcal{I}(Z; Y) \tag{2}$$

where $\beta > 0$. They used (2) as a regularizer to train $f$, minimizing $\mathcal{I}(X; Z)$ to resist MIAs while maximizing $\mathcal{I}(Z; Y)$ to preserve the model's utility.

---

[1]Some literature refers to *hard-label* as *label-only*, and *soft-label* as *black-box*.

## 4 METHODOLOGY

### 4.1 CONDITIONAL MUTUAL INFORMATION-BASED DEFENSE

We aim to resist black-box MIAs where attackers cannot access $Z$, so we still minimize $\mathcal{I}(X;\hat{Y})$ instead of $\mathcal{I}(X;Z)$. Furthermore, we observe that all MIA algorithms target one fixed label during attacking. Formally, let

$$D^y := \{x \in \mathbb{X} : (x,y) \in D\}$$

be the sub-dataset labeled with the ground truth $y$. When $y$ is given, all attackers aim to reconstruct $\hat{D}^y$ as close to $D^y$ as possible. Against their intention, we propose to minimize

$$\mathcal{I}(X;\hat{Y}|Y=y) := \sum_{x \in \mathbb{X}} \sum_{\hat{y} \in \mathbb{Y}} \mathcal{P}(x,\hat{y}|y) \log \frac{\mathcal{P}(x,\hat{y}|y)}{\mathcal{P}(x|y)\mathcal{P}(\hat{y}|y)}. \tag{3}$$

$\mathcal{I}(X;\hat{Y}|Y=y)$ quantifies the dependence between $X$ and $\hat{Y}$ when $Y=y$. We minimize (3) to prevent attackers from inferring the information about $D^y$. To protect the complete $D$, we minimize (3) for each $y \in \mathbb{Y}$ with the weight of $\mathcal{P}(y)$. This is equivalent to minimizing the conditional mutual information (CMI), which is defined as

$$\mathcal{I}(X;\hat{Y}|Y) := \sum_{y \in \mathbb{Y}} \mathcal{P}(y) \cdot \mathcal{I}(X;\hat{Y}|Y=y). \tag{4}$$

**Theorem 1.** *CMI is a special case of Information Bottleneck (2) taking $Z = \hat{Y}$ and $\beta = 1$, i.e.*

$$\mathcal{I}(X;\hat{Y}|Y) = \mathcal{I}(X;\hat{Y}) - \mathcal{I}(\hat{Y};Y).$$

Our proof is in Appendix A. Our theorem proves that CMI inherits the benefits of Information Bottleneck (IB), i.e., minimizing CMI has two effects:

- Minimize $\mathcal{I}(X;\hat{Y})$ to compress the redundant information in inputs, and decrease the dependence between inputs and predictions. This improves the MIA resistance as shown by Wang et al. (2020).
- Maximize $\mathcal{I}(\hat{Y};Y)$ to preserve the useful information for tasks, and increase the dependence between predictions and ground truths. This improves the model's utility obviously.

$\mathcal{I}(X;Z)$ in IB is challenging to calculate because $\mathbb{X}$ and $\mathbb{Z}$ are both high-dimensional. Wang et al. (2020) could only approximate IB by variational bounds (Alemi et al., 2017). Fortunately, as a special case of IB, CMI can be calculated directly (Yang et al., 2024).

### 4.2 MINIMIZE CMI VIA POST-PROCESSING

Previous works minimized CMI during training (Yang et al., 2024; Hamidi et al., 2024; Yang et al., 2025). In contrast to them, we propose a training-free method to minimize CMI. We have

$$\mathcal{I}(X;\hat{Y}|Y) = \sum_{y \in \mathbb{Y}} \mathcal{P}(y) \sum_{x \in \mathbb{X}} \sum_{\hat{y} \in \mathbb{Y}} \mathcal{P}(x,\hat{y}|y) \log \frac{\mathcal{P}(x,\hat{y}|y)}{\mathcal{P}(x|y)\mathcal{P}(\hat{y}|y)}, \qquad \text{by definitions (3-4),}$$

$$= \sum_{x \in \mathbb{X}} \sum_{\hat{y} \in \mathbb{Y}} \sum_{y \in \mathbb{Y}} \mathcal{P}(x,\hat{y},y) \log \frac{\mathcal{P}(\hat{y}|x,y)}{\mathcal{P}(\hat{y}|y)},$$

$$= \sum_{x \in \mathbb{X}} \mathcal{P}(x) \sum_{y \in \mathbb{Y}} \mathcal{P}(y|x) \sum_{\hat{y} \in \mathbb{Y}} \mathcal{P}(\hat{y}|x,y) \log \frac{\mathcal{P}(\hat{y}|x,y)}{\mathcal{P}(\hat{y}|y)},$$

$$= \sum_{x \in \mathbb{X}} \mathcal{P}(x) \sum_{y \in \mathbb{Y}} \mathcal{P}(y|x) \sum_{\hat{y} \in \mathbb{Y}} \mathcal{P}(\hat{y}|x) \log \frac{\mathcal{P}(\hat{y}|x)}{\mathcal{P}(\hat{y}|y)}, \qquad \text{by Markov chain } Y \to X \to \hat{Y}.$$

Thus minimizing $\mathcal{I}(X;\hat{Y}|Y)$ is equivalent to minimizing $\sum_{y \in \mathbb{Y}} \mathcal{P}(y|x) \sum_{\hat{y} \in \mathbb{Y}} \mathcal{P}(\hat{y}|x) \log \frac{\mathcal{P}(\hat{y}|x)}{\mathcal{P}(\hat{y}|y)}$ for each $x$ input to $f$. For simplicity, we sample $y \in \mathbb{Y}$ with the probability of $\mathcal{P}(y|x)$ and minimize

$\sum_{\hat{y} \in \mathbb{Y}} \mathcal{P}(\hat{y}|x) \log \frac{\mathcal{P}(\hat{y}|x)}{\mathcal{P}(\hat{y}|y)}$ instead,[2] which equals the original objective in terms of expectation. Next we need to calculate $\mathcal{P}(\hat{y}|x)$, $\mathcal{P}(y|x)$ and $\mathcal{P}(\hat{y}|y)$.

To get $\mathcal{P}(\hat{y}|x)$, we have $\mathcal{P}(\hat{y}|x) = f_{\hat{y}}(x)$ by design of neural classifiers.

To get $\mathcal{P}(y|x)$, an intuitive idea is that $\mathcal{P}(y|x) = \mathcal{P}(\hat{y}|x)$ for $y = \hat{y}$, but Guo et al. (2017) had demonstrated its inaccuracy in modern neural classifiers. Inspired by their work, we introduce the temperature to calibrate it.

To get $\mathcal{P}(\hat{y}|y)$, we have

$$\mathcal{P}(\hat{y}|y) = \sum_{x \in \mathbb{X}} \mathcal{P}(x, \hat{y}|y) = \sum_{x \in \mathbb{X}} \mathcal{P}(x|y)\mathcal{P}(\hat{y}|x, y) = \sum_{x \in \mathbb{X}} \mathcal{P}(x|y)\mathcal{P}(\hat{y}|x),$$
$$= \sum_{x \in \mathbb{X}} \mathcal{P}(x|y) f_{\hat{y}}(x) = \mathbb{E}_X[f_{\hat{y}}(X) \mid Y = y] \approx \operatorname*{mean}_{x' \in D^y} f_{\hat{y}}(x'),$$

where $\approx$ is by that the samples in $D^y$ are i.i.d. to $\mathcal{P}(x|y)$, and thus the sample mean can estimate the conditional expectation. In practice we use the validation set as $D^y$, because the training samples are overfitted by $f$ and may cause inaccurate estimation.

Now our objective becomes

$$\sum_{\hat{y} \in \mathbb{Y}} \mathcal{P}(\hat{y}|x) \log \frac{\mathcal{P}(\hat{y}|x)}{\mathcal{P}(\hat{y}|y)} \approx \sum_{\hat{y} \in \mathbb{Y}} f_{\hat{y}}(x) \log \frac{f_{\hat{y}}(x)}{\operatorname*{mean}_{x' \in D^y} f_{\hat{y}}(x')} = \mathrm{KL}(\boldsymbol{f}(x) \| \operatorname*{mean}_{x' \in D^y} \boldsymbol{f}(x')), \quad (5)$$

where KL is Kullback-Leibler divergence. To minimize (5), we fix $\operatorname*{mean}_{x' \in D^y} \boldsymbol{f}(x')$ for simplicity and modify $\boldsymbol{f}(x)$. Let $\boldsymbol{p} \in \Delta^{\mathbb{Y}}$ be the modified version of $\boldsymbol{f}(x)$ and our objective is $\mathrm{KL}(\boldsymbol{p} \| \operatorname*{mean}_{x' \in D^y} \boldsymbol{f}(x'))$. Additionally, we constrain $\|\boldsymbol{p} - \boldsymbol{f}(x)\|_1 \leq \varepsilon$ to preserve the model's utility, where $\varepsilon > 0$ is the distortion controller.

In information theory, minimizing the mutual information under bounded distortion constraints is known as the rate-distortion problem (Shannon, 1959), which is for signal compression. If a signal has less information, it is easier to compress, and a stricter distortion bound can be applied. Inspired by his work, we introduce Shannon entropy to quantify the information in $\hat{Y}$ when $X = x$, which is defined as

$$\mathcal{H}(\hat{Y}|X = x) := -\sum_{\hat{y} \in \mathbb{Y}} \mathcal{P}(\hat{y}|x) \log \mathcal{P}(\hat{y}|x).$$

Our constraint becomes $\|\boldsymbol{p} - \boldsymbol{f}(x)\|_1 \leq \varepsilon \cdot \mathcal{H}(\hat{Y}|X = x)$, where the distortion bound is proportional to the amount of information. This new constraint reduces the modification when the information is limited, and enhances the compression when the information is abundant. We refer to this as *adaptive rate-distortion*.

Our defense is summarized as Algorithm 1. Without the need to retrain the model, our defense is plug-and-play and easy to deploy.

Note that $D' := \{\boldsymbol{q}^y : y \in \mathbb{Y}\}$ can be calculated and stored in advance. If the model owner and the defender are not the same, the owner only needs to provide $D'$ instead of $D$, avoiding communication costs and privacy risks.

(6) can be solved by existing convex optimizers. Furthermore, we derive the explicit solution in Appendix B, calculate it within $O(|\mathbb{Y}| \log |\mathbb{Y}|)$ time in Algorithm 2, accelerate it via GPUs in Algorithm 3, and evaluate the computational cost in Appendix C.

---

[2] With sampling, we only need to consider one $y \in \mathbb{Y}$, so we can solve (6) within $O(|\mathbb{Y}| \log |\mathbb{Y}|)$ time by Algorithm 2. Without sampling, we have to consider all $y \in \mathbb{Y}$. The problem complexity is $\Omega(|\mathbb{Y}|^2)$, which is unacceptable when $\mathbb{Y}$ is large.

---

**Algorithm 1:** post-processing to minimize CMI.

---

**Input:** original output $\boldsymbol{f}(x)$, validation set $D$, temperature $T$, distortion controller $\varepsilon$.
**Output:** modified output $\boldsymbol{p}^*$.
Sample $y \in \mathbb{Y}$ with the probability of **softmax**$(\frac{\boldsymbol{f}(x)}{T})$;
$\boldsymbol{q}^y \leftarrow \underset{x' \in D^y}{\textbf{mean}} \boldsymbol{f}(x')$;
$\mathcal{H} \leftarrow - \underset{\hat{y} \in \mathbb{Y}}{\sum} f_{\hat{y}}(x) \log f_{\hat{y}}(x)$;
Solve the convex optimization problem:

$$
\begin{aligned}
\min \quad & \mathrm{KL}(\boldsymbol{p} \| \boldsymbol{q}^y), \\
\text{s.t.} \quad & \|\boldsymbol{p} - \boldsymbol{f}(x)\|_1 \leq \varepsilon \cdot \mathcal{H}, \\
& \boldsymbol{p} \in \Delta^{\mathbb{Y}}.
\end{aligned} \tag{6}
$$

**return** the optimal solution $\boldsymbol{p}^*$;

---

## 5 EXPERIMENTS

### 5.1 SETTINGS

**Datasets.** We select CelebA (Liu et al., 2015) and FaceScrub (Ng & Winkler, 2014) as the private datasets, and FFHQ (Karras et al., 2018) as the public dataset. CelebA has 10,177 labels and we take 1,000 labels with the most images. FaceScrub has 530 labels and 106,863 images, but we only take 43,147 images because the other URLs are broken. FFHQ has 70,000 unlabeled images. All images are cropped and resized to $64 \times 64$. We use 80% of the private samples for training, 10% for validation, and 10% for testing.

**Models.** We select IR-152 (He et al., 2015) and VGG-16 (Simonyan & Zisserman, 2014) as the target models, and MaxViT (Tu et al., 2022) as the evaluation models. IR-152 and MaxViT are pre-trained on MS-Celeb-1M (Guo et al., 2016), and VGG-16 is pre-trained on ImageNet (Deng et al., 2009). They are fine-tuned on the training set, and we select the best version by the validation set. The evaluation model on CelebA achieves 97.3% test accuracy, and the one on FaceScrub achieves 99.3%.

**Attacks.** We select Mirror (An et al., 2022), C2FMI (Ye et al., 2023) and RLBMI (Han et al., 2023) as the soft-label attackers, and BREP (Kahla et al., 2022) and LOKT (Nguyen et al., 2023) as the hard-label attackers. They attack the first 100 private labels and reconstruct 5 images per label. For RLBMI, BREP and LOKT, we train GANs and surrogate models on FFHQ. For Mirror and C2FMI, we use the $256 \times 256$ StyleGAN2 (Karras et al., 2019) trained on FFHQ, whose generated images are center-cropped to $176 \times 176$ and resized to $64 \times 64$.

**Defenses.** We select MID (Wang et al., 2020), BiDO (Peng et al., 2022), LS (Struppek et al., 2023), TL (Ho et al., 2024) and Purifier (Yang et al., 2023) as the competitors. Purifier trains a CVAE, and the others retrain the target models. We carefully tune their hyper-parameters to achieve the similar validation accuracies. All hyper-parameters of defenses are in Appendix D.

To evaluate the MIA resistance and model's utility, we consider the following metrics:

**Attack Accuracy** Let the evaluation model reclassify the reconstructed images. The top-1 and top-5 accuracies are denoted as $Acc1$ and $Acc5$ respectively. Lower percentages indicate better MIA resistance.

**Feature Distance** The image features are extracted from the penultimate layer of a model. We take the average $L_2$ feature distance between the reconstructed images and the nearest training images. The features are extracted by the evaluation model and a FaceNet (Schroff et al., 2015) trained on VGGFace2 (Cao et al., 2017), denoted as $\delta_{eval}$ and $\delta_{face}$ respectively. Higher distances indicate better MIA resistance.

**Test Accuracy** The accuracy of the target model on the test set, denoted as $Acc$. Higher percentage indicates better utility.

**Distortion** The $L_1$ distance between the outputs with and without defense. We take the average on the test set, denoted as $Dist$. Lower distance indicates better utility.

All experiments are conducted on MIBench (Qiu et al., 2024b). The experiments about RLBMI, Purifier and high resolution are in Appendix E-G.

Table 1: MIA resistance of various defenses under soft-label attacks.

| | | Mirror | | | | C2FMI | | | |
|---|---|---|---|---|---|---|---|---|---|
| | | $\downarrow Acc1$ | $\downarrow Acc5$ | $\uparrow \delta_{eval}$ | $\uparrow \delta_{face}$ | $\downarrow Acc1$ | $\downarrow Acc5$ | $\uparrow \delta_{eval}$ | $\uparrow \delta_{face}$ |
| IR-152 CelebA | None | 18.0% | 31.0% | 625 | 1.22 | 5.8% | 15.4% | 647 | 1.30 |
| | MID | 17.8% | 31.6% | 629 | 1.22 | **1.2%** | 4.2% | 699 | 1.48 |
| | BiDO | 10.6% | 25.8% | 614 | 1.16 | 5.4% | 13.0% | 645 | 1.33 |
| | LS | 9.0% | 15.2% | 660 | 1.36 | 1.8% | 8.0% | 660 | 1.36 |
| | TL | 13.6% | 28.6% | 633 | 1.24 | 2.8% | 5.0% | 703 | 1.48 |
| | SSD | **3.2%** | **8.2%** | **728** | **1.46** | **1.2%** | **1.8%** | **744** | **1.59** |
| IR-152 FaceScrub | None | 55.2% | 76.8% | 496 | 0.89 | 19.6% | 41.4% | 549 | 1.02 |
| | MID | 38.0% | 61.0% | 534 | 0.96 | **0.2%** | **1.8%** | **715** | **1.44** |
| | BiDO | 34.4% | 60.4% | 526 | 0.93 | 13.2% | 27.6% | 588 | 1.13 |
| | LS | 45.8% | 73.4% | 503 | 0.90 | 13.6% | 31.2% | 564 | 1.05 |
| | TL | 39.2% | 63.2% | 535 | 0.99 | 7.0% | 18.6% | 617 | 1.22 |
| | SSD | **32.2%** | **46.4%** | **604** | **1.15** | 1.6% | 5.6% | 671 | 1.35 |
| VGG-16 FaceScrub | None | 29.0% | 51.8% | 544 | 1.00 | 12.8% | 30.6% | 588 | 1.11 |
| | MID | 34.6% | 64.8% | 520 | 0.94 | 4.4% | 17.8% | 611 | 1.19 |
| | BiDO | 20.0% | 44.4% | 556 | 1.03 | 12.0% | 27.6% | 575 | 1.09 |
| | LS | 30.2% | 56.4% | 531 | 0.97 | 15.6% | 37.4% | 551 | 1.05 |
| | TL | 17.8% | 41.8% | 558 | 1.03 | 8.4% | 27.2% | 574 | 1.10 |
| | SSD | **16.4%** | **40.0%** | **604** | **1.13** | **3.2%** | **13.6%** | **656** | **1.29** |

Table 2: MIA resistance of various defenses under hard-label attacks.

| | | BREP | | | | LOKT | | | |
|---|---|---|---|---|---|---|---|---|---|
| | | $\downarrow Acc1$ | $\downarrow Acc5$ | $\uparrow \delta_{eval}$ | $\uparrow \delta_{face}$ | $\downarrow Acc1$ | $\downarrow Acc5$ | $\uparrow \delta_{eval}$ | $\uparrow \delta_{face}$ |
| IR-152 CelebA | None | 26.8% | 50.8% | 541 | 1.02 | 50.2% | 67.8% | 527 | 0.93 |
| | MID | 20.8% | 39.4% | 596 | 1.13 | 32.4% | 52.2% | 551 | 1.00 |
| | BiDO | 11.4% | 33.4% | 577 | 1.09 | 30.8% | 48.4% | 576 | 1.06 |
| | LS | 19.4% | 43.8% | 547 | 1.03 | 33.6% | 57.0% | 537 | 1.01 |
| | TL | 20.2% | 46.4% | 550 | 1.05 | 43.0% | 66.2% | 525 | 0.93 |
| | SSD | **1.8%** | **4.2%** | **769** | **1.62** | **1.2%** | **9.0%** | **769** | **1.61** |
| IR-152 FaceScrub | None | 50.6% | 77.2% | 497 | 0.95 | 90.2% | 96.2% | 411 | 0.65 |
| | MID | 36.2% | 52.6% | 570 | 1.10 | 61.8% | 81.4% | 475 | 0.78 |
| | BiDO | 33.6% | 62.6% | 530 | 1.02 | 82.2% | 94.8% | 445 | 0.69 |
| | LS | 36.0% | 67.0% | 521 | 1.01 | 84.8% | 95.4% | 429 | 0.69 |
| | TL | 30.4% | 59.0% | 538 | 1.06 | 91.6% | 98.4% | 414 | 0.65 |
| | SSD | **9.2%** | **13.4%** | **717** | **1.47** | **15.8%** | **25.0%** | **656** | **1.27** |
| VGG-16 FaceScrub | None | 28.2% | 54.0% | 543 | 1.06 | 75.8% | 91.4% | 444 | 0.74 |
| | MID | 30.0% | 56.8% | 535 | 1.03 | 63.2% | 86.8% | 470 | 0.80 |
| | BiDO | 21.6% | 44.4% | 566 | 1.10 | 67.4% | 87.4% | 474 | 0.80 |
| | LS | 24.0% | 50.4% | 549 | 1.05 | 74.4% | 90.6% | 453 | 0.79 |
| | TL | 20.8% | 46.8% | 566 | 1.08 | 67.4% | 88.8% | 471 | 0.79 |
| | SSD | **10.8%** | **18.6%** | **693** | **1.40** | **24.0%** | **35.4%** | **635** | **1.24** |

## 5.2 COMPARISONS WITH STATE-OF-THE-ART DEFENSES

Table 1 shows the evaluation results under soft-label attacks, and Table 2 shows the ones under hard-label. Our SSD exhibits the best MIA resistance and outperforms all state-of-the-art defenses, across various attack algorithms, private datasets, and model architectures. This is because we minimize the CMI to prevent attackers from inferring the information about the private dataset.

Surprisingly, some defenses perform worse than no defense in some indicators. It demonstrates that existing defenses may fail to resist modern black-box attacks. They focus on modifying the weights and structure of the model, but black-box attackers only exploit the outputs and thus are rarely hindered. This highlights the necessity of designing specialized black-box defenses.

Figure 2 shows the reconstructed images from IR-152 models trained on CelebA. Our SSD significantly enhances the difference between $D^y$ and $\hat{D}^y$, forces attackers to reconstruct wrong images, and thus protects the private data. In contrast, other defenses fail to prevent attackers from reconstructing similar images, leading to privacy leakage.

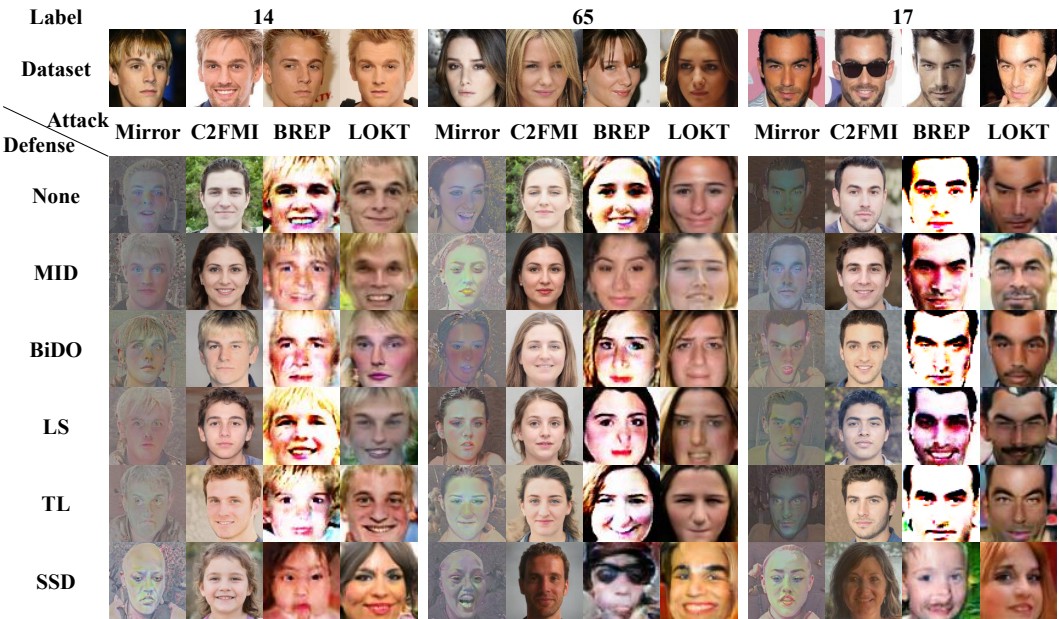

Figure 2: The reconstructed images from various attacks facing various defenses. Above are the ground truth labels $y$ and the private sub-datasets $D^y$ (4 images shown). Below are the reconstructed datasets $\hat{D}^y$ (1 image shown).

Table 3: Model's utility with various defenses.

|  | IR-152 & CelebA | | IR-152 & FaceScrub | | VGG-16 & FaceScrub | |
|---|---|---|---|---|---|---|
|  | $\uparrow Acc$ | $\downarrow Dist$ | $\uparrow Acc$ | $\downarrow Dist$ | $\uparrow Acc$ | $\downarrow Dist$ |
| **None** | 92.1% | 0 | 98.2% | 0 | 91.8% | 0 |
| **MID** | 86.8% | 0.607 | 95.5% | 0.321 | 86.4% | 0.744 |
| **BiDO** | 86.6% | 0.371 | 95.3% | 0.135 | 88.5% | 0.313 |
| **LS** | 86.9% | 0.317 | 95.6% | 0.103 | 87.5% | 0.295 |
| **TL** | 86.5% | 0.352 | 95.8% | 0.128 | 87.4% | 0.306 |
| **SSD** | **87.1%** | **0.191** | **97.0%** | **0.055** | **89.3%** | **0.176** |

Table 3 shows the evaluation results on model's utility. Our SSD achieves the highest test accuracy, because we minimize CMI to preserve the information useful for tasks. Additionally, SSD holds the lowest distortion because of our adaptive rate-distortion constraint.

## 5.3 VISUALIZE THE OUTPUTS

We use t-SNE (van der Maaten & Hinton, 2008) to visualize how SSD modifies the outputs. The target model is a VGG-16 model trained on FaceScrub. Figure 3 shows that minimizing CMI makes the outputs aggregate into clusters, as demonstrated by Yang et al. (2024). For private samples, this reduces the sensitivity between inputs and outputs, preventing attackers from inferring the private information. For attack samples, this misleads attackers towards the wrong label.

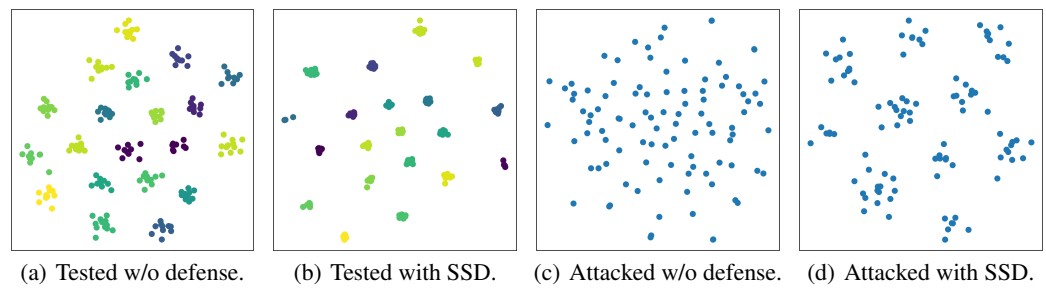

(a) Tested w/o defense. (b) Tested with SSD. (c) Attacked w/o defense. (d) Attacked with SSD.

Figure 3: Visualizing the outputs by t-SNE. For (a) and (b), the test samples are colored according to their ground truth labels. For (c) and (d), the attack samples are generated by Mirror.

## 5.4 ABLATION EXPERIMENTS

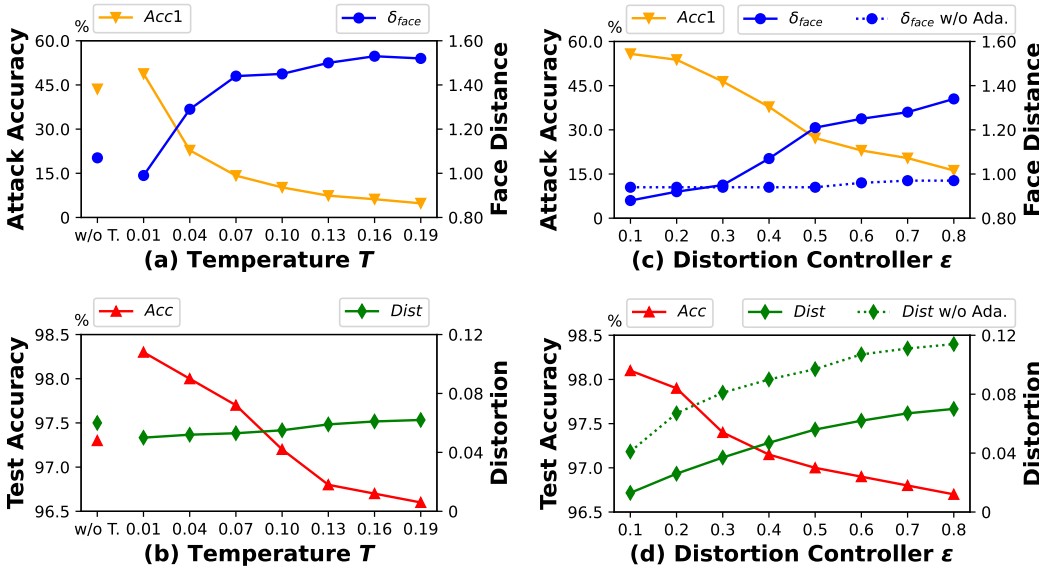

Figure 4: The ablation results. The attacker of (a) is BREP, and (c) is Mirror. "w/o T." denotes "without temperature", and "w/o Ada." denotes "without adaptive rate-distortion".

To explore the effects of temperature $T$ and distortion controller $\varepsilon$, we conduct ablation experiments on IR-152 models trained on FaceScrub.

Figure 4(a) shows that a higher temperature helps to resist hard-label attacks, because it makes the sampling probability more uniform, and attackers easier to get misleading labels. However, a higher temperature decreases the test accuracy, as shown in Figure 4(b). Specifically, without the temperature, neither the MIA resistance nor model's accuracy is satisfactory, which highlights the necessity of temperature.

Figure 4(c) shows that a larger distortion bound helps to resist soft-label attacks, because it allows greater modifications. However, a larger distortion bound decreases the test accuracy and increases the distortion, as shown in Figure 4(d). Specifically, without the adaptive rate-distortion, neither the face distance nor the distortion is satisfactory, which highlights the necessity of adaption.

## 6 Conclusion

In contrast to previous researches on model inversion defense with a focus on white-box attacks, we conduct a specific study on black-box attacks. Specifically, we investigate the impact of conditional mutual information (CMI) and develop a CMI-based defense strategy. We conduct our defense in the post-processing stage instead of re-training the model. To further reduce the modifications to outputs, we introduce an adaptive rate-distortion framework and optimize it by the water-filling method. Experimental results demonstrate that our defense method achieves state-of-the-art (SOTA) performance against black-box attacks. We hope that our findings will help shift attention toward robust defense mechanisms in black-box settings and inspire further researches in this area.

## 7 Acknowledgement

This work is supported by National Natural Science Foundation of China (62171248, 62301189), Peng Cheng Laboratory (PCL2023A08), and Shenzhen Science and Technology Program (KJZD20240903103702004, JCYJ20220818101012025, RCBS20221008093124061, GXWD20220811172936001).

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

## A    PROOF OF THEOREM 1

$$\mathcal{I}(X; \hat{Y}|Y),$$

$$= \sum_{y \in \mathbb{Y}} \mathcal{P}(y) \sum_{x \in \mathbb{X}} \sum_{\hat{y} \in \mathbb{Y}} \mathcal{P}(x, \hat{y}|y) \log \frac{\mathcal{P}(x, \hat{y}|y)}{\mathcal{P}(x|y)\mathcal{P}(\hat{y}|y)}, \qquad \text{by definitions (3-4),}$$

$$= \sum_{x \in \mathbb{X}} \sum_{\hat{y} \in \mathbb{Y}} \sum_{y \in \mathbb{Y}} \mathcal{P}(x, \hat{y}, y) \log \frac{\mathcal{P}(\hat{y}|x, y)}{\mathcal{P}(\hat{y}|y)},$$

$$= \sum_{x \in \mathbb{X}} \sum_{\hat{y} \in \mathbb{Y}} \sum_{y \in \mathbb{Y}} \mathcal{P}(x, \hat{y}, y) \log \frac{\mathcal{P}(\hat{y}|x)}{\mathcal{P}(\hat{y}|y)}, \qquad \text{by Markov chain } Y \to X \to \hat{Y},$$

$$= \sum_{x \in \mathbb{X}} \sum_{\hat{y} \in \mathbb{Y}} \sum_{y \in \mathbb{Y}} \mathcal{P}(x, \hat{y}, y) \log \left( \frac{\mathcal{P}(x, \hat{y})}{\mathcal{P}(x)} \middle/ \frac{\mathcal{P}(\hat{y}, y)}{\mathcal{P}(y)} \right),$$

$$= \sum_{x \in \mathbb{X}} \sum_{\hat{y} \in \mathbb{Y}} \sum_{y \in \mathbb{Y}} \mathcal{P}(x, \hat{y}, y) \log \left( \frac{\mathcal{P}(x, \hat{y})}{\mathcal{P}(x)\mathcal{P}(\hat{y})} \middle/ \frac{\mathcal{P}(\hat{y}, y)}{\mathcal{P}(\hat{y})\mathcal{P}(y)} \right),$$

$$= \sum_{x \in \mathbb{X}} \sum_{\hat{y} \in \mathbb{Y}} \mathcal{P}(x, \hat{y}) \log \frac{\mathcal{P}(x, \hat{y})}{\mathcal{P}(x)\mathcal{P}(\hat{y})} - \sum_{\hat{y} \in \mathbb{Y}} \sum_{y \in \mathbb{Y}} \mathcal{P}(\hat{y}, y) \log \frac{\mathcal{P}(\hat{y}, y)}{\mathcal{P}(\hat{y})\mathcal{P}(y)},$$

$$= \mathcal{I}(X; \hat{Y}) - \mathcal{I}(\hat{Y}; Y), \qquad \text{by definition (1).}$$

## B    WATER-FILLING TO SOLVE (6)

For brevity, let $\boldsymbol{q} := \boldsymbol{q}^y$, $\boldsymbol{f} := \boldsymbol{f}(x)$, and $\varepsilon := \varepsilon \cdot \mathcal{H}$. Then (6) is restated as

$$\begin{aligned} \min \quad & \mathrm{KL}(\boldsymbol{p}\|\boldsymbol{q}), \\ \text{s.t.} \quad & \|\boldsymbol{p} - \boldsymbol{f}\|_1 \le \varepsilon, \\ & \boldsymbol{p} \in \Delta^{\mathbb{Y}}. \end{aligned} \qquad (7)$$

Let $\boldsymbol{p}^*$ be the optimal solution. Note that $\mathrm{KL}(\boldsymbol{p}\|\boldsymbol{q})$ is a convex function with the minimum point $\boldsymbol{p} = \boldsymbol{q}$. We have $\boldsymbol{p}^* = \boldsymbol{q}$ if $\|\boldsymbol{q} - \boldsymbol{f}\|_1 \le \varepsilon$, and $\|\boldsymbol{p}^* - \boldsymbol{f}\|_1 = \varepsilon$ otherwise.

Consider $\|\boldsymbol{q} - \boldsymbol{f}\|_1 > \varepsilon$ in the following. Since $\boldsymbol{p}^*, \boldsymbol{f} \in \Delta^{\mathbb{Y}}$, we have

$$\sum_{i \in \mathbb{Y}:\, p_i^* > f_i} |p_i^* - f_i| = \sum_{i \in \mathbb{Y}:\, p_i^* < f_i} |p_i^* - f_i| = \frac{\varepsilon}{2}. \qquad (8)$$

**Lemma 1.** $\forall i \in \mathbb{Y}$, either $q_i \ge p_i^* \ge f_i$ or $q_i \le p_i^* \le f_i$.

*Proof.* Assume that $\exists i \in \mathbb{Y}, p_i^* > q_i \ge f_i$. Since $\boldsymbol{p}^*, \boldsymbol{q} \in \Delta^{\mathbb{Y}}$, there must $\exists j \in \mathbb{Y}, p_j^* < q_j$. Let

$$g(\xi) := (p_i^* - \xi) \log \frac{p_i^* - \xi}{q_i} + (p_j^* + \xi) \log \frac{p_j^* + \xi}{q_j}.$$

Note that $g'(\xi)$ is continuous and $g'(0) = \log \frac{p_j^*}{q_j} - \log \frac{p_i^*}{q_i} < 0$, so $\exists \xi^* > 0, \forall \xi \in (0, \xi^*)$, $g(\xi) < g(0)$. This implies that decreasing $p_i^*$ by $\xi$ and increasing $p_j^*$ by $\xi$ will reduce $\mathrm{KL}(\boldsymbol{p}^*\|\boldsymbol{q})$, contradicting the optimality of $\boldsymbol{p}^*$. No constraints are violated if $\xi$ is chosen such that $p_i^* - \xi \ge f_i$ and $p_j^* + \xi \le 1$. $\square$

Let $A := \{i \in \mathbb{Y} : q_i > f_i\}$ and $B := \mathbb{Y} \setminus A$. Based on (8) and Lemma 1, we divide (7) into two sub-problems, (9) and (10).

$$\begin{aligned} \min \quad & \sum_{i \in A} p_i \log \frac{p_i}{q_i}, \\ \text{s.t.} \quad & \sum_{i \in A} p_i - f_i = \frac{\varepsilon}{2}, \\ & q_i \ge p_i \ge f_i, \qquad i \in A. \end{aligned} \quad (9) \qquad \begin{aligned} \min \quad & \sum_{i \in B} p_i \log \frac{p_i}{q_i}, \\ \text{s.t.} \quad & \sum_{i \in B} p_i - f_i = -\frac{\varepsilon}{2}, \\ & q_i \le p_i \le f_i, \qquad i \in B. \end{aligned} \quad (10)$$

To solve (9), we introduce Lagrange multipliers $v^* \in \mathbb{R}$ for $\sum_{i \in A} p_i^* - f_i = \frac{\varepsilon}{2}$, and $\lambda_i^* \geq 0$ for $p_i^* \geq f_i$, $i \in A$. The KKT conditions are

$$1 + \log \frac{p_i^*}{q_i} - v^* - \lambda_i^* = 0,$$
$$(p_i^* - f_i)\lambda_i^* = 0.$$

Eliminating $\lambda_i^* \geq 0$ yields

$$1 + \log \frac{p_i^*}{q_i} \geq v^*, \tag{11}$$

$$(p_i^* - f_i)\left(1 + \log \frac{p_i^*}{q_i} - v^*\right) = 0. \tag{12}$$

If $1 + \log \frac{f_i}{q_i} \geq v^*$, then $p_i^* > f_i$ implies $1 + \log \frac{p_i^*}{q_i} > v^*$ contradicting (12), so $p_i^* = f_i$.

If $1 + \log \frac{f_i}{q_i} < v^*$, then (11) implies $p_i^* > f_i$, and (12) implies $p_i^* = q_i \exp(v^* - 1)$.

In summary,

$$p_i^* = \begin{cases} f_i, & 1 + \log \frac{f_i}{q_i} \geq v^*, \\ q_i \exp(v^* - 1), & \text{other.} \end{cases} \tag{13}$$

Let $w_A^* := \exp(v^* - 1)$ and (13) becomes

$$p_i^* = \max(f_i, w_A^* q_i), \quad i \in A, \tag{14}$$

where $w_A^*$ satisfies $\sum_{i \in A} p_i^* - f_i = \frac{\varepsilon}{2}$. Let $c(w) := \sum_{i \in A} \max(0, wq_i - f_i)$, which is a piecewise-linear increasing function with breakpoints $w = \frac{f_i}{q_i}$, $i \in A$. Note that $c(0) = 0$ and $c(1) = \frac{\|q - f\|_1}{2} > \frac{\varepsilon}{2}$, we can find $w_A^* \in (0, 1)$ such that $c(w_A^*) = \frac{\varepsilon}{2}$. The constraints $q_i \geq p_i^*$, $i \in A$ are naturally satisfied by $w_A^* < 1$.

Solving (10) similarly yields

$$p_i^* = \min(f_i, w_B^* q_i), \quad i \in B, \tag{15}$$

where $w_B^*$ satisfies $\sum_{i \in B} p_i^* - f_i = -\frac{\varepsilon}{2}$ and $w_B^* \in (1, +\infty)$.

Combining (14-15) and $w_A^* < 1 < w_B^*$, the solution of (7) is

$$p_i^* = \min(\max(f_i, w_A^* q_i), w_B^* q_i), \quad i \in \mathbb{Y}. \tag{16}$$

We propose Algorithm 2 to solve (7) efficiently. Firstly we address the trivial case. For non-trivial cases, we ensure $\frac{f_1}{q_1} \leq \frac{f_2}{q_2} \leq \ldots \leq \frac{f_{|\mathbb{Y}|}}{q_{|\mathbb{Y}|}}$ by sorting. To find $w_A^*$, we check $w = \frac{f_1}{q_1}, \frac{f_2}{q_2}, \ldots$ successively. Once $c(w) > \frac{\varepsilon}{2}$ at $w = \frac{f_j}{q_j}$, we get $w_A^* = \frac{\frac{\varepsilon}{2} + \sum_{i=1}^{j-1} f_i}{\sum_{i=1}^{j-1} q_i} \in [\frac{f_{j-1}}{q_{j-1}}, \frac{f_j}{q_j})$. This process is known as *water-filling*, because $w$ is like a rising water level, $\frac{f_1}{q_1}, \frac{f_2}{q_2}, \ldots$ are like ascending steps, and $\frac{\varepsilon}{2}$ is like the total volume of water. To find $w_B^*$, we check $w = \frac{f_{|\mathbb{Y}|}}{q_{|\mathbb{Y}|}}, \frac{f_{|\mathbb{Y}|-1}}{q_{|\mathbb{Y}|-1}}, \ldots$ similarly. This symmetric process is known as *reverse water-filling*. Finally we undo the sorting and return (16). Our time complexity is $O(|\mathbb{Y}| \log |\mathbb{Y}|)$ due to the sorting.

Furthermore, we propose Algorithm 3 to utilize GPU acceleration. Leveraging the PyTorch operators, we manage to eliminate the **if** and **while** statements in Algorithm 2, making Algorithm 3 sequential completely and suitable for GPUs. Algorithm 3 can solve many $(f, q)$ parallelly in one batch, achieving lower computational costs.

**Algorithm 2:** CPU-based water-filling.

**Input:** $f_i, q_i$ for $i \in \mathbb{Y}$.
**Output:** $p_i^*$ for $i \in \mathbb{Y}$.
**if** $\sum_{i \in \mathbb{Y}} |q_i - f_i| \leq \varepsilon$ **then**
  | **return** $q_i$ for $i \in \mathbb{Y}$;
Reindex $f_i, q_i$ so that $\frac{f_1}{q_1} \leq \frac{f_2}{q_2} \leq \ldots \leq \frac{f_{|\mathbb{Y}|}}{q_{|\mathbb{Y}|}}$;
$j \leftarrow 1$;
$F \leftarrow 0$;
$Q \leftarrow 0$;
**while** $\frac{f_j}{q_j} Q - F \leq \frac{\varepsilon}{2}$ **do**
  | $F \leftarrow F + f_j$;
  | $Q \leftarrow Q + q_j$;
  | $j \leftarrow j + 1$;
$w_A^* \leftarrow \dfrac{F + \frac{\varepsilon}{2}}{Q}$;
$j \leftarrow |\mathbb{Y}|$;
$F \leftarrow 0$;
$Q \leftarrow 0$;
**while** $\frac{f_j}{q_j} Q - F \geq -\frac{\varepsilon}{2}$ **do**
  | $F \leftarrow F + f_j$;
  | $Q \leftarrow Q + q_j$;
  | $j \leftarrow j - 1$;
$w_B^* \leftarrow \dfrac{F - \frac{\varepsilon}{2}}{Q}$;
Restore the indices of $f_i, q_i$;
**return** $\min(\max(f_i, w_A^* q_i), w_B^* q_i)$ for $i \in \mathbb{Y}$;

---

**Algorithm 3:** GPU-based water-filling.

**Input:** PyTorch tensors $\boldsymbol{f}, \boldsymbol{q}$ of size $|\mathbb{Y}|$.
**Output:** PyTorch tensor $\boldsymbol{p}^*$ of size $|\mathbb{Y}|$.
$m \leftarrow (\|\boldsymbol{q} - \boldsymbol{f}\|_1 \leq \varepsilon)$;

Reindex $\boldsymbol{f}, \boldsymbol{q}$ by torch.sort($\frac{\boldsymbol{f}}{\boldsymbol{q}}$);

$\boldsymbol{F} \leftarrow \boldsymbol{f}$.cumsum();
$\boldsymbol{Q} \leftarrow \boldsymbol{q}$.cumsum();
$\boldsymbol{M} \leftarrow (\frac{\boldsymbol{f}}{\boldsymbol{q}} \boldsymbol{Q} - \boldsymbol{F} \leq \frac{\varepsilon}{2})$;

$j \leftarrow \boldsymbol{M}$.argmin();
$w_A^* \leftarrow \dfrac{\boldsymbol{F}[j] + \frac{\varepsilon}{2}}{\boldsymbol{Q}[j]}$;

$\boldsymbol{F} \leftarrow 1 - \boldsymbol{F}$;
$\boldsymbol{Q} \leftarrow 1 - \boldsymbol{Q}$;
$\boldsymbol{M} \leftarrow (\frac{\boldsymbol{f}}{\boldsymbol{q}} \boldsymbol{Q} - \boldsymbol{F} \geq -\frac{\varepsilon}{2})$;

$j \leftarrow \boldsymbol{M}$.argmax();
$w_B^* \leftarrow \dfrac{\boldsymbol{F}[j] - \frac{\varepsilon}{2}}{\boldsymbol{Q}[j]}$;
Restore the indices of $\boldsymbol{f}, \boldsymbol{q}$;
**return** $\boldsymbol{q}$.where($m$, $\boldsymbol{f}$.clip($w_A^* \boldsymbol{q}, w_B^* \boldsymbol{q}$));

## C  COMPUTATIONAL COST

We evaluate the time cost of Algorithm 1, whose convex optimization problem is solved by Algorithm 3. We take a batch with 512 images and infer 100 times. The time cost is measured by torch.profiler, an official tool provided by PyTorch. We exclude the time for I/O (i.e. from disk to memory, and from CPU to GPU), only counting the time for computation. Our experiment is conducted on a NVIDIA GeForce RTX 3090.

Table 4: Time cost of Algorithm 1 equipped with Algorithm 3.

|  | IR-152 & CelebA | IR-152 & FaceScrub | VGG-16 & FaceScrub |
|---|---|---|---|
| **None** | 18.63 s | 17.70 s | 5.65 s |
| **SSD** | 19.22 s | 18.16 s | 6.07 s |
| Increment | 3.1% | 2.5% | 7.4% |

Table 4 shows that we increase little time. The higher percentage on VGG-16 is due to the shallower model architecture. In absolute terms, modifying 512 outputs for 100 times only needs 0.5 seconds. If we take the I/O time into account, the percentages will be low enough to be ignored.

## D  HYPER-PARAMETERS OF DEFENSES

Table 5: Hyper-parameters of the defenses in our experiments.

|  | IR-152 & CelebA | IR-152 & FaceScrub | VGG-16 & FaceScrub |
|---|---|---|---|
| **MID** | $\beta = 0.005$ | $\beta = 0.02$ | $\beta = 0.015$ |
| **BiDO** | $\lambda_x = 0.0005, \lambda_y = 0.005$ | $\lambda_x = 0.005, \lambda_y = 0.05$ | $\lambda_x = 0.0005, \lambda_y = 0.005$ |
| **LS** | $\alpha = -0.1$ | $\alpha = -0.05$ | $\alpha = -0.05$ |
| **TL** | freeze 50% layers | freeze 50% layers | freeze 50% layers |
| **SSD** | $T = 0.1, \varepsilon = 0.35$ | $T = 0.1, \varepsilon = 0.5$ | $T = 0.1, \varepsilon = 0.8$ |

## E  UNDER RLBMI ATTACK

Table 6: MIA resistance of various defenses under RLBMI attack.

|  | $\downarrow Acc1$ | $\downarrow Acc5$ | $\uparrow \delta_{eval}$ | $\uparrow \delta_{face}$ |
|---|---|---|---|---|
| **None** | 44% | 65% | 505 | 0.89 |
| **MID** | 31% | 53% | 523 | 0.90 |
| **BiDO** | 24% | 44% | 552 | 1.01 |
| **LS** | 24% | 52% | 527 | 0.98 |
| **TL** | 37% | 55% | 523 | 0.94 |
| **SSD** | **21%** | **42%** | **572** | **1.07** |

We conduct the experiment under RLBMI (Han et al., 2023), a soft-label attack. Aligned with Tables 1-3, the target models are IR-152 trained on CelebA. Due to the huge computational cost of RLBMI, we only reconstruct 1 image per label. Table 6 shows that our SSD still outperforms the other defenses.

## F  COMPARISON WITH PURIFIER

Table 7: Comparisons between Purifier and SSD.

|  |  | $\downarrow Acc1$ | $\downarrow Acc5$ | $\uparrow \delta_{eval}$ | $\uparrow \delta_{face}$ |
|---|---|---|---|---|---|
| Mirror | **Purifier** | 3.6% | **8.2%** | 721 | 1.43 |
|  | **SSD** | **3.2%** | **8.2%** | **728** | **1.46** |
| C2FMI | **Purifier** | **0.6%** | 2.6% | 723 | 1.56 |
|  | **SSD** | 1.2% | **1.8%** | **744** | **1.59** |
| BREP | **Purifier** | 22.8% | 41.6% | 591 | 1.14 |
|  | **SSD** | **1.8%** | **4.2%** | **769** | **1.62** |
| LOKT | **Purifier** | 40.2% | 57.6% | 546 | 0.99 |
|  | **SSD** | **1.2%** | **9.0%** | **769** | **1.61** |
|  |  | $\uparrow Acc$ | $\downarrow Dist$ |  |  |
| Utility | **Purifier** | 84.5% | 0.362 |  |  |
|  | **SSD** | **87.1%** | **0.191** |  |  |

Purifier (Yang et al., 2023) is a black-box defense against membership inference attacks, and perhaps resists model inversion attacks. Despite the lack of details about $\lambda$ and $k$NN, we reproduce their

work setting $\lambda = k = 1$. If the $L_2$ distance between the input and the nearest training sample is less than 0.00005, then we swap the top-1 and top-2 labels. The validation set is used to train the CVAE. Aligned with Tables 1-3, the target model is the same IR-152 trained on CelebA. Table 7 shows that our SSD outperforms Purifier.

## G    HIGH RESOLUTION

We use HD-CelebA-Cropper[3] to generate high resolution images, which are cropped and resized to $224 \times 224$. The IR-152 and MaxViT models are retrained on this new CelebA, and the test accuracy of MaxViT achieves 97.2%.

We select Mirror as the attacker, equipped with the $1024 \times 1024$ StyleGAN2 trained on FFHQ. The generated images are center-cropped to $800 \times 800$ and resized to $224 \times 224$. Due to the huge computational cost for high resolution, we only attack the first 20 labels and reconstruct 5 images per label. Table 8 shows that our SSD still outperforms the other defenses.

Table 8: Comprehensive results on high resolution.

| | Mirror | | | | Utility | | Hyper-parameter |
|---|---|---|---|---|---|---|---|
| | $\downarrow Acc1$ | $\downarrow Acc5$ | $\uparrow \delta_{eval}$ | $\uparrow \delta_{face}$ | $\uparrow Acc$ | $\downarrow Dist$ | |
| **None** | 32% | 53% | 492 | 1.25 | 96.4% | 0 | |
| **MID** | 37% | 68% | 477 | 1.28 | 93.1% | 0.334 | $\beta = 0.005$ |
| **BiDO** | 37% | 55% | 487 | 1.23 | 94.7% | 0.112 | $\lambda_x = 0.05, \lambda_y = 0.5$ |
| **LS** | 28% | 43% | 508 | 1.29 | 94.1% | 0.118 | $\alpha = -0.005$ |
| **TL** | 64% | 96% | 472 | 1.17 | 94.1% | 0.719 | freeze 80% layers |
| **SSD** | **17%** | **33%** | **522** | **1.30** | **95.0%** | **0.079** | $T = 0.1, \varepsilon = 1.5$ |

## H    DISCUSSION ON ADAPTIVE ATTACKS

A potential adaptive attack strategy is:

1. Query the same $x$ repeatedly and count the frequency of different outputs.
2. Estimate our sampling probability by the frequency.
3. Infer our original output by the sampling probability.

If an online server detects such pattern of queries, it can block them. Step back and consider again, we propose a memory-free and low-cost strategy to block such adaptive attack:

Design a hash function $h \colon \mathbb{X} \to \mathbb{N}$ where $\mathbb{N}$ is the integer set. When users/attackers query $x$, we take $h(x)$ as the random seed for sampling, ensuring same-input-same-output. Locality Sensitive Hash (Gionis et al., 1999) can be used to cope with the perturbations to $x$.

---

[3]https://github.com/LynnHo/HD-CelebA-Cropper

