# OpenReview forum: "Stealthy Shield Defense: A Conditional Mutual Information-Based Approach against Black-Box Model Inversion Attacks"
_ICLR.cc/2025/Conference — ICLR 2025 Poster_

### Official Review · Reviewer_J1L4 · 2024-10-29

**Soundness:** 4
**Presentation:** 3
**Contribution:** 3
**Rating:** 6
**Confidence:** 4

**Summary:**

This paper addresses the critical issue of protecting machine learning models from black-box model inversion attacks (MIAs). Specifically, the attacker’s objective is to reconstruct private training data by only accessing the model's outputs.The authors therefore propose a novel defense mechanism named Stealthy Shield Defense (SSD) to post-process the model’s output such that the information revealed by the model’s output about the private data is minimized under constrain. This method leverages Conditional Mutual Information (CMI) to reduce the dependency between the model's output and private data. In addition, they also propose an adaptive rate-distortion framework using water-filling method to preserve the utility while minimizing the CMI.

**Strengths:**

1. The authors propose SSD, a post-processing defense mechanism that doesn’t require retraining the model, making it practical for real  deployment.

2. The authors theoretically prove that minimizing CMI serves as a special information bottleneck, therefore minimizing CMI can effectively balance data privacy and utility. By iterating the CMI through all possible labels, the whole dataset can thus be.

3. The paper introduces an adaptive rate-distortion mechanism optimized using the water-filling method. This approach efficiently calibrates the probability distributions output by the model.

4. The authors validate their method across various attack algorithms (BREP, Mirror, C2F, LOKT), datasets (FaceScrub, CelebA), and model architectures (VGG-16, IR-152). The results demonstrate that SSD outperforms existing defenses in terms of defense scheme robustness and effectiveness on preservation of model utility.

**Weaknesses:**

1. Even though the authors claim that the computational overhead is negligible due to the efficient optimization on GPU, a more detailed analysis or benchmarking of the computational cost would greatly support this claim.

2. Using MI/CMI in deep learning is gaining increasing attention. However, its introduction in related work lacks both depth and breadth, which makes it hard to find the role of this work into the relevant community.

3. The authors should use a deeper model architecture and higher-dimensional input data for training and prove the effectiveness of the proposed method. When the input data is in high dimensionality, it usually contains a significant amount of irrelevant information. Even worse, since the model depth is also high, the final output, Y ̂, may only contain a small amount of MI with the input, X. I am wondering if optimizing I(X;Y ̂ ∣ Y) will be challenging in this case.

4. Even though this is a robust scheme against model inversion attack, the authors should discuss about the potential possibility of adaptive attacks. If the adaptive attack is unlikely to happen for now, the authors should also state the reason why.

**Questions:**

1. The proposed scheme requires iterating over all possible labels to perform the defense. What happens if there are a large number of possible labels? For example, a large company or region may train a face recognition model that includes thousands or millions of faces. Will this scale slow down the defense and become a bottleneck? I suggest some detailed computational analysis.

2. As the authors pointed out in the paper, X→Z→Y ̂ is a Markov chain. It is possible that when input data has large dimensionality, or the model has very deep layers. Under this circumstance, it is likely that Y ̂ shares very little information with X. The derived MI I(X;Y ̂ ∣ Y) will have such a small value that could be hardly optimized. Does the proposed scheme still hold effective in this scenario? The authors should provide more empirical/theoretical analysis to justify this.

3. Are there any manifest adaptive attack schemes that can target at this defense? How difficult is it to design/launch such attacks?

---

> ### Author Response · Authors · 2024-11-25
>
> We sincerely express our gratitude for dedicating your valuable time to providing positive comments that will enhance our paper. Your praise regarding our applicability, insight, methodology and experiments, have greatly encouraged and motivated us. Our detailed responses to all of your conerns are presented below.
>
>     Even though the authors claim that the computational overhead is negligible due to the efficient optimization on GPU, a more detailed analysis or benchmarking of the computational cost would greatly support this claim.
>
> Following your suggestion, we add a quantitative experiment stated in [Common Concerns on Computational Cost](https://openreview.net/forum?id=p0DjhjPXl3&noteId=aptS4M72M9), where the results indicate that **modifying 512 predictions for 100 times only needs 0.5 seconds with our careful optimization.**
>
>     The proposed scheme requires iterating over all possible labels to perform the defense. What happens if there are a large number of possible labels? For example, a large company or region may train a face recognition model that includes thousands or millions of faces. Will this scale slow down the defense and become a bottleneck? I suggest some detailed computational analysis.
>
> As analyzed [above](https://openreview.net/forum?id=p0DjhjPXl3&noteId=gp8hRQ4nt2), we have simplified the objective function, reducing the space complexity from $\Theta(|\mathbb{Y}|^2)$ to $\Theta(|\mathbb{Y}|)$, and time complexity from $\Theta(|\mathbb{Y}|^2)$ to $\Theta(|\mathbb{Y}|\log|\mathbb{Y}|)$.
>
> Furthermore, we add an experiment to illustrate the relationship between $|\mathbb{Y}|$ and time-cost. We generate $\pmb{x}\in\mathbb{R}^{|\mathbb{Y}|}\sim N(\pmb{0},\pmb{I})$ and let $\pmb{y}=\text{softmax}(10\cdot\pmb{x})$. It is observed that the $\pmb{y}$ generated in this way is close to the real probability distributions. We generate $\pmb{f}(x)$ and $\pmb{q}^y$ randomly in this way, and let our GPU-based-water-filling to find its optimal solution $\pmb{p}$. We take a batch with 256 pairs $(\pmb{f}(x),\pmb{q}^y)$. We record the time by `torch.profiler`, an official tool provided by PyTorch. Our experiment is conducted on one NVIDIA GeForce RTX 3090. Here is the result:
>
> | $\|\mathbb{Y}\|$ | $10^1$ | $10^2$ | $10^3$ | $10^4$ | $10^5$ | $10^6$ |
> | -------------- | :----: | :----: | :----: | :----: | :----: | :----: |
> |    Time    |  131 ms  |  132 ms   |  143 ms   |  163 ms   |  249 ms   |  1301 ms  |
>
> It shows that **even when $|\mathbb{Y}|$ reaches a million, solving 256 convex optimization problems only takes us 1.3 seconds.** We believe that at this point, our post-processing will not be the performance bottleneck, but the slow inferring and massive parameters of the target model will be.
>
>     Using MI/CMI in deep learning is gaining increasing attention. However, its introduction in related work lacks both depth and breadth, which makes it hard to find the role of this work into the relevant community.
>
> Thank you for pointing this out. The application of conditional mutual information (CMI) to deep learning was proposed this year [1]. So far, there are only three published works on this topic. In the revised version of our paper, we will supplement mutual information (MI) and information bottlenecks (IB) in related work.
>
>     Even though this is a robust scheme against model inversion attack, the authors should discuss about the potential possibility of adaptive attacks. If the adaptive attack is unlikely to happen for now, the authors should also state the reason why.
>
>     Are there any manifest adaptive attack schemes that can target at this defense? How difficult is it to design/launch such attacks?
>
> Yes! Your question has prompted us to think deeply about adaptive attacks. In [Common Concerns on Adaptive Attacks](https://openreview.net/forum?id=p0DjhjPXl3&noteId=gjDthbJkrW), we discuss the adaptive attack strategies that may apply to us, and our countermeasures against them.
>
> ---
> [1] En-Hui Yang, Shayan Mohajer Hamidi, Linfeng Ye, Renhao Tan, and Beverly Yang. Conditional mutual information constrained deep learning: Framework and preliminary results. IEEE International Symposium on Information Theory, 2024.

---

> > ### Comment · Reviewer_J1L4 · 2024-11-26
> >
> > Thank you for responding to my previous comments. While most of my concerns are addressed, I notice that there was no response to one of them. I have specifically raised the concern that testing the proposed method on deeper model architectures and higher-dimensional input data is necessary. In high-dimensional settings, input data often contains a significant amount of irrelevant information, which can make optimizing I(X; Ŷ ∣ Y) more challenging. Moreover, as model depth increases, the final output, Ŷ, may retain only a small amount of MI with the input X. I am curious about how the proposed method performs under these conditions, as this scenario is more representative of real-world applications with complex data. Could you provide clarification on this?

---

> ### Author Response · Authors · 2024-11-26
>
> This is indeed a profound and pivotal question. Since the first day we saw it, we have been thinking about it. Now, we can finally provide a satisfactory clarification on this, both theoretically and experimentally.
>
> ---
>
> **Theoretically, higher-dimensional $X$ lead to a higher conditional mutual information $\mathbb{I}(X;\hat{Y}|Y)$, not lower.**
> As you said, higher-dimensional $X$ contains more information ***irrelevant*** to $\hat{Y}$, but it also contains more information ***relevant*** to $\hat{Y}$. Our CMI only quantifies the relevant parts (by its definition). The more relevant information, the greater the CMI.
>
> Furthermore, we provide a mathematical proof. Let $X_{high}$ be the high-dimensional input, $X_{low}$ be the low-dimensional input, and $X_{diff}$ be the difference between them. There is
> $$
> \mathbb{I}(X_{high};\hat{Y}|Y)
> $$
> $$
> =\mathbb{I}(X_{low},X_{diff};\hat{Y}|Y)\qquad\qquad\qquad\qquad\quad(1)
> $$
> $$
> =\mathbb{I}(X_{low};\hat{Y}|Y)+\mathbb{I}(X_{diff};\hat{Y}|Y,X_{low})\qquad(2)
> $$
> $$
> \geq\mathbb{I}(X_{low};\hat{Y}|Y)\qquad\qquad\qquad\qquad\qquad\qquad(3)
> $$
> where (2) is based on the chain rule of MI, and (3) is based on the non-negativity of MI. Now we have proven that $X_{high}$ has a higher CMI than $X_{low}$.
>
> ---
>
> **Theoretically, deeper neural networks lead to a lower conditional mutual information $\mathbb{I}(X;\hat{Y}|Y)$, as you said.** Without considering the residual connections, neural networks can be regarded as a Markov chain:
> $$Y \rightarrow X \rightarrow Z_{1} \rightarrow Z_{2} \rightarrow \ldots \rightarrow Z_{d} \rightarrow \hat{Y}$$
> where $Y$ is the true label, $X$ is the input, $Z_{1},Z_{2},...,Z_{d}$ are intermediate features, $\hat{Y}$ is the predicted label, and $d$ is the depth of the neural network. Based on the ***data processing inequality [1]***, there is
> $$\mathbb{I}(X;Z_{1})\geq\mathbb{I}(X;Z_{2})\geq\ldots\geq\mathbb{I}(X;Z_{d})\geq\mathbb{I}(X;\hat{Y})$$
> which means that MI decreases as the neural network becomes deeper. We can prove that CMI also has a similar inequality:
> $$\mathbb{I}(X;Z_{1}|Y)\geq\mathbb{I}(X;Z_{2}|Y)\geq\ldots\geq\mathbb{I}(X;Z_{d}|Y)\geq\mathbb{I}(X;\hat{Y}|Y)$$
> which means that CMI decreases as the neural network becomes deeper. So your concern is justified!
>
> Fortunately, there are two factors that prevent CMI from being too small to optimize:
>
> 1. **Deeper networks are often used for higher-dimensional inputs in practice.** As we have demonstrated above, higher-dimensional $X$ brings higher CMI, offsetting the negative impact of deeper models.
> 2. **Deeper networks have residual connections,** which disrupt the Markov property. Their Markov chains should be changed into
> $$Y \rightarrow X \rightarrow (Z_{1},Z_{2},\ldots,Z_{d}) \rightarrow \hat{Y}$$
> which greatly shortens the length of the data processing inequality.
>
> ---
>
> To verify our theoretical analysis, we conducted quantitative experiments. We select models with different depths and train them on FaceSrube with different resolutions. We calculated their CMI on the test set, and the results are:
>
> | Experiment | Resolution |   Model    |  CMI  |                  Comment                   |
> | :--------: | :--------: | :--------: | :---: | :----------------------------------------: |
> |     A      |  224×224   | ResNet-152 | 0.232 |  High-dimensional Inputs, Deep Model   |
> |     B      |   64×64    |   IR-152   | 0.215 |   Low-dimensional Inputs, Deep Model   |
> |     C      |   64×64    |   VGG-16   | 0.227 | Low-dimensional Inputs, Shallow Model  |
>
> We can see that:
> - CMI is higher in A than in B, confirming our theory that higher-dimensional inputs lead to a higher CMI.
> - CMI is lower in B than in C, confirming our theory that deeper neural networks lead to a lower CMI.
> - The three CMIs differ little and are on the same order of magnitude, indicating that CMIs are numerically stable and **will not be too small to optimize**.
>
> In fact, the MIA robustness and model's utility of Experiment A are shown in [High resolution and RLB results](https://openreview.net/forum?id=p0DjhjPXl3&noteId=4YN1OTkglv), and the ones of Experiment B and C are shown in Tables 1-3 in our paper, indicating that **our defense hold effective in various input dimension and model depth**.
>
> ---
>
> Thank you again for asking such a profound question! In the process of thinking and responding, we realized that MI/CMI might be the key to explaining the effectiveness of residual connections! This is certainly worth a further special research. We will try to explain deep learning via information theory in the future.
>
> If you have any more questions, please feel free to continue asking us.
>
> ---
>
> [1] https://en.wikipedia.org/wiki/Data_processing_inequality

---

### Official Review · Reviewer_KsSS · 2024-10-30

**Soundness:** 3
**Presentation:** 2
**Contribution:** 2
**Rating:** 6
**Confidence:** 3

**Summary:**

This paper proposes a post-processing based defense to protect against black-box model inversion attack. The key insight of the paper is to transform the output of the model to reduce conditional mutual information (CMI). This is done to reduce the dependence between the output and the input while preserving the dependence on the true labels. The authors develop a mechanism to transform the model’s prediction to reduce CMI by framing it as an optimization problem. The experimental results on multiple image classification datasets show that their proposed SSD defense provides a better privacy-utility tradeoff compared to prior defenses.

**Strengths:**

1. The paper is well-written and easy to follow.
2. The idea behind the proposed defense is intuitive and presented well.
3. Experiments seem comprehensive and the defense shows better utility-privacy tradeoff compared to prior defenses in both soft and hard label settings
4. The defense is post-processing based, making it easy to adopt.

**Weaknesses:**

1. The proposed defense could be susceptible to an adaptive attack. An adversary could query the same input multiple times to obtain multiple predictions from the model. Since the defense produces outputs by perturbing the original prediction, the adversary could compute an average over multiple outputs to get a better estimate of the model’s true output. Such an adaptive attack is not discussed by the paper.
2. The defense requires a validation dataset to implement, which could limit its adoption.

**Questions:**

1. In line 228, the authors state that “the objective function is too complex for the convex optimizer to solve.” and use this as the motivation to minimize a simplified objective $KL(p||q^y)$ instead (by sampling $y\in\mathbb{Y}$). Why is this the original objective too complex? Wouldn’t you be able to use gradient descent to solve for $p$? Would this lead to a better solution?
2. How was the temperature $T$ picked in Algorithm 1?
3. What was the value of $\epsilon$ for the proposed defense in the experiments?
4. Why is the Acc@1 for IR-152 lower for “no defense” compared to prior defenses? This suggests that adding prior defense improves the attack success rate, which is very strange.
5. Can you address the adaptive attack discussed in Weakness#1?

---

> ### Author Response · Authors · 2024-11-23
>
> Thank you for careful review and positive feedback! We are honored that you appreciate our writing, ideas, experiments and applicability.
>
> Below are our point-by-point responses to your questions.
>
>     The proposed defense could be susceptible to an adaptive attack. An adversary could query the same input multiple times to obtain multiple predictions from the model. Since the defense produces outputs by perturbing the original prediction, the adversary could compute an average over multiple outputs to get a better estimate of the model’s true output. Such an adaptive attack is not discussed by the paper.
>
>     Can you address the adaptive attack discussed in Weakness#1?
>
> Thank you for pointing out an interesting point of adaptive attacks, which has inspired our deeper thinking. Our detailed discussion and solution is stated in [Common Concerns on Adaptive Attacks](https://openreview.net/forum?id=p0DjhjPXl3&noteId=gjDthbJkrW). We will supplement the discussion in our paper.
>
>     The defense requires a validation dataset to implement, which could limit its adoption.
>
>
> We conducted additional experiments (as stated [above](https://openreview.net/forum?id=p0DjhjPXl3&noteId=amDzrfS2C9)), and our conclusion is: **if the model owner has not prepared a validation set, using the training set is also acceptable.** This will not affect the robustness of MIA, but only slightly affect the model's utility.
>
>     In line 228, the authors state that “the objective function is too complex for the convex optimizer to solve.” and use this as the motivation to minimize a simplified objective $\mathbb{KL}(\pmb{p}||\pmb{q}^{y})$ instead (by sampling $y\in\mathbb{Y}$). Why is this the original objective too complex? Wouldn’t you be able to use gradient descent to solve for? Would this lead to a better solution?
>
> The original optimization objective
> $$\sum_{y\in\mathbb{Y}}{\mathbb{P}(y|\pmb{x})\sum_{\hat{y}\in\mathbb{Y}}{\mathbb{P}(\hat{y}|\pmb{x})\log\frac{\mathbb{P}(\hat{y}|\pmb{x})}{\mathbb{P}(\hat{y}|y)}}}$$
>
> is too complex because it involves $|\mathbb{Y}|$ vectors of length $|\mathbb{Y}|$, **resulting in a space complexity and time complexity of $\Theta(|\mathbb{Y}|^2)$, which is unacceptable.** After sampling $y\in\mathbb{Y}$, the simplified objective only involves one vector of length $|\mathbb{Y}|$, with a space complexity of $\Theta(|\mathbb{Y}|)$ and a time complexity of $\Theta(|\mathbb{Y}|\log|\mathbb{Y}|)$ for sorting, which is acceptable.
>
> Convex optimization problems can indeed be solved by gradient descent, which is the approach taken by most famous optimizers. **However, the computational cost of gradient descent is unacceptable** because we require a solution for each prediction. Based on the simplified objective, we are able to propose water-filling to solve it, and efficiently implement it on GPUs. More details are stated in [Common Concerns on Computational Cost](https://openreview.net/forum?id=p0DjhjPXl3&noteId=aptS4M72M9).
>
> **In terms of mathematical expectation, our simplified form is equivalent to the original form.** We sample based on probability $\mathbb{P}(y|\pmb{x})$, while the original form is a probability-weighted-sum. However, we introduce randomness, vulnerable to adaptive attacks as you mentioned. Our discussion and solution are stated in [Common Concerns on Adaptive Attacks](https://openreview.net/forum?id=p0DjhjPXl3&noteId=gjDthbJkrW)
>
>     How was the temperature $T$ picked in Algorithm 1?
>
>     What was the value of $\varepsilon$ for the proposed defense in the experiments?
>
> All hyperparameters are listed in Table 4 in our paper, which is
>
> |               | IR-152 & CelebA | IR-152 & FaceScrub | VGG-16 & FaceScrub |
> | :-----------: | :-------------: | :----------------: | :----------------: |
> |      $T$      |     $0.03$      |       $0.05$       |       $0.10$       |
> | $\varepsilon$ |       $1$       |        $1$         |        $1.5$         |
>
>     Why is the Acc@1 for IR-152 lower for “no defense” compared to prior defenses? This suggests that adding prior defense improves the attack success rate, which is very strange.
>
> That's exactly what happened. Previous defenses have become ineffective against the latest and advanced attacks, and may even facilitate them. This also highlights the importance of our work. In the Table 5 of the latest MIA survey [1], there are similar results.
>
> ---
>
> Thank you again for your recognition! If you have any other questions, please feel free to ask us.
>
> ---
>
> [1] Qiu, Yixiang et al. “MIBench: A Comprehensive Benchmark for Model Inversion Attack and Defense.” ArXiv abs/2410.05159 (2024): n. pag.

---

### Official Review · Reviewer_Ui99 · 2024-11-03

**Soundness:** 3
**Presentation:** 3
**Contribution:** 3
**Rating:** 6
**Confidence:** 4

**Summary:**

This paper proposes a post-processing defense SSD based on conditional mutual information (CMI) especially for defending against black-box model inversion attacks. It theoretically proves that CMI is effective and further reduce modifications to outputs by proposing adaptive rate-distortion framework. Experiment results indicate that SSD achieves SOTA better utility-privacy trade-off.

**Strengths:**

-	This paper considers defense against model inversion attack from a new perspective by proposing CMI, which provides new insights into this field.
-	It provides thorough theoretical analysis to support the effectiveness of the SSD defense, with a clear and rigorous logical structure.
-	The paper achieves state-of-the-art performance in terms of utility-privacy trade-off and defense against advanced black-box attacks, qualitatively demonstrating its effectiveness.

**Weaknesses:**

-	Key concepts, such as the water-filling method, lack adequate coverage in the main body, making it challenging for readers to fully grasp the methodology without delving into appendices. Greater emphasis on these aspects within the main text would improve the paper's accessibility.
-	The experimental setup omits certain critical information, such as the dataset used for GAN prior training and specifics for Figure 1, leaving questions regarding reproducibility and completeness.
-	The study exclusively benchmarks against state-of-the-art white-box defenses but omits comparisons with black-box defenses, which would be more pertinent for the black-box MIA context.


Minor remarks:

-	There are also minor errors, such as duplicated words (line 107), inconsistent notation (duplicate use of $p$ at line 215) and typo at line 314.
-	The notation is unconventional, such as using non-bold font for vector inputs like $\mathbf{x}$, which reduces consistency with standard notation practices.

**Questions:**

-	How does SSD perform when the distribution of the public dataset differs significantly from the private dataset (e.g., public dataset = FFHQ, private dataset = CelebA)? Clarification on this point would help to understand SSD’s adaptability under various deployment conditions.
-	The paper states that SSD shows exceptional performance in reducing attack accuracy for LOKT and BREP methods. Could the authors elaborate on why SSD is particularly effective against these specific methods?
-	Could the authors provide additional details on the GAN prior training and any specific hyperparameters used for Figure 1?

---

> ### Author Response · Authors · 2024-11-22
>
> We would like to express our gratitude for your valuable suggestions and meaningful feedback. We are honored that you consider us to provide new insights into MIA field, with a rigorous theoretical structure and excellent experimental results.
>
> Our detailed responses are provided below.
>
>     Key concepts, such as the water-filling method, lack adequate coverage in the main body, making it challenging for readers to fully grasp the methodology without delving into appendices. Greater emphasis on these aspects within the main text would improve the paper's accessibility.
>
> Thanks for the advice. We fully agree with you. We will carefully revise our paper, placing the optimal solution of water-filling in the main text. Due to the page limitations, the mathematical derivation of water-filling has to be placed in the appendix.
>
>     The experimental setup omits certain critical information, such as the dataset used for GAN prior training and specifics for Figure 1, leaving questions regarding reproducibility and completeness.
>
>     Could the authors provide additional details on the GAN prior training and any specific hyperparameters used for Figure 1?
>
> **All our GANs are trained on the public dataset FFHQ. For each attack method, our GAN training is consistent with the original paper.** Specifically, for Mirror and C2F, we use StyleGAN2-ADA [1], a well-known pre-trained GAN. Its weights can be downloaded on [github](https://github.com/NVlabs/stylegan2-ada-pytorch). For BREP, we follow the [official implementation](https://github.com/AI-secure/GMI-Attack/blob/master/Celeba/train_gan.py) to train a GAN, changing the public dataset from CelebA to FFHQ. For LOKT, we generate pseudo-labels by the target model (with each defense) and train an ACGAN [2]. The trainning details are consistent with the [official code](https://github.com/ngoc-nguyen-0/LOKT_neurips2023/blob/main/train_tacgan.py).
>
> The detailed settings for Figures 1~2 are presented in Section 5.3. We use the VGG-16 trained on FaceScrub as the target model, consistent with Tables 1-3. Our defense hyperparameters are $T=0.1, ε=1.5$ as presented in Table 4. We employ `TSNE` implemented by `sklearn.manifold`, where we set `metric=jensenshannon` and all other parameters to be default. The CMI is calculated by Equation (2) in the paper.
>
> Thank you again for pointing out these details. Your suggestions will greatly improve the quality of our paper. If our paper is accepted, **we will open all code so that readers will have no concerns about reproducibility.**
>
>     The study exclusively benchmarks against state-of-the-art white-box defenses but omits comparisons with black-box defenses, which would be more pertinent for the black-box MIA context.
>
> **Our competitors, MID, BiDO, LS and TL, can all be considered as special black-box defenses.** Because they not only modify the model's weights, but also the model's outputs (including soft-labels and hard-labels). In fact, in the Table 8 of [3], the authors of LS conducted experiments on defending against black-box attacks, indicating that LS is designed for both white-box and black-box scenarios.
>
> To our knowledge, our SSD is the first defense specifically designed for black-box scenario. Our paper thoroughly demonstrates the necessity and significance behind this. Our competitors, MID, BiDO, LS and TL, are all SOTA defenses (and naturally, SOTA black-box defenses as well). **According to the latest MIA survey [4], we have compared with all the competitors that should be compared.**
>
> ---
> [1] Tero Karras, Miika Aittala, Janne Hellsten, Samuli Laine, Jaakko Lehtinen, and Timo Aila. Training generative adversarial networks with limited data. In Proc. NeurIPS, 2020.
>
> [2] Augustus Odena, Christopher Olah, and Jonathon Shlens. Conditional image synthesis with auxiliary classifier gans. In International conference on machine learning, pages 2642–2651. PMLR, 2017.
>
> [3] Struppek, Lukas et al. “Be Careful What You Smooth For: Label Smoothing Can Be a Privacy Shield but Also a Catalyst for Model Inversion Attacks.” ArXiv abs/2310.06549 (2023): n. pag.
>
> [4] Qiu, Yixiang et al. “MIBench: A Comprehensive Benchmark for Model Inversion Attack and Defense.” ArXiv abs/2410.05159 (2024): n. pag.

---

> > ### Comment · Reviewer_Ui99 · 2024-11-23
> >
> > Thank you for your response. You have partially addressed my concerns, but I believe there are still some **conceptual misunderstandings**, as outlined below:
> >
> > - **white-box attacks** allow access to model weights during the attack, whereas **black-box attacks** are limited to accessing soft or hard labels. White-box attacks are inherently more harmful as they exploit more information about the model.
> >
> > - **white-box defenses** (e.g., MID, BiDO, LS, and TL) enhance a model’s robustness against white-box attacks by modifying the model’s weights during training.  Since these methods enhance robustness against white-box attacks, they inherently lead to improved defense capabilities against black-box attacks as well, which are less powerful. (**Re: MID, BiDO, LS and TL can all be considered as special black-box defenses;  LS is designed for both white-box and black-box scenarios.**) In contrast, **black-box defenses** are designed solely to defend against black-box attacks, as they do not modify the model’s training process or weights. Therefore, I believe it is not entirely fair to compare your method with white-box defenses.
> >
> > Moreover, **your method is not the first defense specifically designed for the black-box scenario**. It is possible that MIBench overlooked some existing methods; please refer to [1, 2] for further context.
> >
> > I would be pleased to consider increasing the score if you can address this issue effectively.
> >
> >
> > [1] Yang et al, “Purifier: Defending Data Inference Attacks via Transforming Confidence Scores”, AAAI 2023.
> >
> > [2] Wen et al, “Defending Against Model Inversion Attack by Adversarial Examples”.

---

> > > ### Author Response · Authors · 2024-11-23
> > >
> > > Thank you for recommending those two papers. They are not open-sourced, and we are working hard to reproduce them.
> > >
> > > Now we continue to address your other questions.
> > >
> > >     There are also minor errors, such as duplicated words (line 107), inconsistent notation (duplicate use of at line 215) and typo at line 314.
> > >
> > >     The notation is unconventional, such as using non-bold font for vector inputs like, which reduces consistency with standard notation practices.
> > >
> > > Thank you for pointing these out and we will revise carefully. Specifically, we will take your suggestion to use bold font for vectors and non-bold font for scalars.
> > >
> > >     How does SSD perform when the distribution of the public dataset differs significantly from the private dataset (e.g., public dataset = FFHQ, private dataset = CelebA)? Clarification on this point would help to understand SSD’s adaptability under various deployment conditions.
> > >
> > > Exactly as you said, we use FFHQ as public dataset and CelebA as private dataset. Furthermore, we have conducted three experiments in our paper, whose settings are
> > >
> > > | Public Dataset | Private Dataset | Target Model |
> > > | :------------: | :-------------: | :----------: |
> > > |      FFHQ      |     CelebA      |    IR-152    |
> > > |      FFHQ      |    FaceScrub    |    IR-152    |
> > > |      FFHQ      |    FaceScrub    |    VGG-16    |
> > >
> > > These various experiments show that our defense has good applicability and universality.
> > >
> > >     The paper states that SSD shows exceptional performance in reducing attack accuracy for LOKT and BREP methods. Could the authors elaborate on why SSD is particularly effective against these specific methods?
> > >
> > > Both LOKT and BREP are hard-label attacks. Our defense indeed may modify hard labels (just like MID, BiDO, LS and TL). The modifications depend on: (1) Our convex optimization model itself, and (2) our sampling in $\mathbb{Y}$ with probability $\mathbb{P}(y|\pmb{x})$. In our ablation study (Figure 4), we found that the higher temperature $T$, the better MIA robustness against BREP. Because when $T$ is high, the sampling probability $\mathbb{P}(y|\pmb{x})$ approaches a uniform distribution. However, higher $T$ impairs the model's utility, which is also stated in the our ablation experiment.
> > >
> > > From the perspective of information theory, by minimizing the CMI, we reduce the information of $X$ carried by $\hat{Y}$. This naturally applies to hard labels as well. However, as Reviewer KsSS and J1L4 have pointed out, they can launch **adaptive attacks**, where querying the same input repeatedly and taking the average (for soft labels) or the mode (for hard labels). Our discussion and solution on adaptive attacks are stated in [Common Concerns on Adaptive Attacks](https://openreview.net/forum?id=p0DjhjPXl3&noteId=gjDthbJkrW).
> > >
> > > We are working hard to reproduce the works you mentioned. If you have any other questions, please feel free to ask us.

---

> > > ### Author Response · Authors · 2024-12-01
> > > **We Outperform the Two Black-Box Defenses You Mentioned**
> > >
> > > We successfully reproduce the two black-box defenses you mentioned, Purifier [1] and Adversarial [2], despite the lack of open-source code. Here are our details:
> > >
> > > - Purifier [1] needs $D_{ref}$ that has the same distribution with $D_{train}$ but no members. To satisfy this, we re-split FaceScrub, using 80% as $D_{train}$, 10% as $D_{ref}$, 10% as $D_{test}$. The target model is re-trained on new $D_{train}$. Our defense use $D_{train}$ as $D_{valid}$ and not use $D_{ref}$. Note that **this favors Purifier because it uses 10% more data.**
> > > - Purifier [1] lacks details on $\lambda$ and $k$NN. After trying our best, we find the optimal values $\lambda=0.01$ and $k=1$. For a target model's prediction $F(x)$, we find the nearest in $P_{index}$. If their L2 distance $<0.0001$, we swap the top1 and top2 labels. Other settings follow the original paper.
> > > - Adversarial [2] does not modify hard labels, which is outdated and cannot defend against advanced label-only attacks. To enhance its MIA robustness, we remove its distortion and hard-label constraints, retaining probability constraint and maximizing inversion error. **If we don't remove the two constraints, [2] has no defensive effect and the comparison is meaningless.** Other settings follow the original paper, where $iteration=10$ and $step=0.4$.
> > >
> > > The comparisons are aligned with Table 1-4 in our paper. The target model is IR-152 and fixed for each defense. The GANs are trained on FFHQ. The first 100 classes in FaceScrub are attacked and each reconstructs 5 images.
> > >
> > > The MIA robustness against C2F:
> > >
> > > |             | $↓Acc@1$ | $↓Acc@5$ | $↑δ_{eval}$ | $↑δ_{face}$ |
> > > | :---------: | -------: | -------: | ----------: | ----------: |
> > > |    None     |    28.0% |    47.4% |        1949 |        1.01 |
> > > |  Purifier   |     3.7% |     7.9% |    **2655** |        1.47 |
> > > | Adversarial |     7.0% |    13.6% |        2260 |        1.28 |
> > > |  **Ours**   | **1.7%** | **3.5%** |        2518 |    **1.49** |
> > >
> > > The MIA robustness against BREP:
> > >
> > > |             | $↓Acc@1$ | $↓Acc@5$ | $↑δ_{eval}$ | $↑δ_{face}$ |
> > > | :---------: | -------: | -------: | ----------: | ----------: |
> > > |    None     |    37.4% |    54.2% |        2150 |        1.00 |
> > > |  Purifier   |    38.6% |    56.8% |        2140 |        1.00 |
> > > | Adversarial |    28.6% |    47.8% |        2250 |        1.10 |
> > > |  **Ours**   | **2.6%** | **3.7%** |    **2650** |    **1.55** |
> > >
> > > The MIA robustness against Mirror (only 20 classes attacked due to huge time cost of Mirror):
> > >
> > > |             | $↓Acc@1$ | $↓Acc@5$ | $↑δ_{eval}$ | $↑δ_{face}$ |
> > > | :---------: | -------: | -------: | ----------: | ----------: |
> > > |    None     |      67% |      88% |        1752 |        0.77 |
> > > |  Purifier   |      58% |      73% |        1915 |        0.99 |
> > > | Adversarial |      63% |      85% |        1858 |        0.82 |
> > > |  **Ours**   |  **25%** |  **37%** |    **2272** |    **1.21** |
> > >
> > > The target model's utility:
> > >
> > > |             |    $↑Acc$ | $↓\text{Avg }L_1$ | $↓\text{Max }L_1$ |
> > > | :---------: | --------: | ----------------: | ----------------: |
> > > |    None     |     98.4% |                 0 |                 0 |
> > > |             |           |                   |                   |
> > > |  Purifier   |     96.0% |              0.14 |              2.00 |
> > > | Adversarial |     96.1% |              0.22 |              1.83 |
> > > |  **Ours**   | **96.5%** |          **0.06** |          **0.95** |
> > >
> > > **It can be seen that our defense outperforms the two black-box defenses you mentioned, both in MIA robustness and utility.** In addition, here are our comments on them:
> > > - Purifier [1] is specifically designed for member inference attacks, with some defense effect on other attacks. However, **its MIA robustness is relatively weak and cannot compete with ours.**
> > > - Adversarial [2] needs a MIA model $A$ to simulate attackers, which is unrealistic in our view. **Defenders should not assume how attackers will attack them, because attacks can be in any way.** Our defense is based on information theory, which is more realistic.
> > >
> > > Thank you again for pointing out these two papers. We will add the above comparisons and comments on them to our final paper. If you have any other questions, please feel free to continue asking us.
> > >
> > > ---
> > >
> > > [1] Yang et al, “Purifier: Defending Data Inference Attacks via Transforming Confidence Scores”, AAAI 2023.
> > >
> > > [2] Wen et al, “Defending Against Model Inversion Attack by Adversarial Examples”.

---

> > > > ### Comment · Reviewer_Ui99 · 2024-12-01
> > > >
> > > > Thanks for your efforts, and I have increased my score.

---

### Official Review · Reviewer_VqZg · 2024-11-04

**Soundness:** 2
**Presentation:** 2
**Contribution:** 2
**Rating:** 6
**Confidence:** 4

**Summary:**

This paper proposes a post-process black-box MI defense Stealthy Shield Defense (SSD) based on conditional mutual information (CMI). By leveraging CMI, SSD aims to reduce the model's output dependence on its input. The experiments demonstrate SSD's effectiveness against four state-of-the-art black-box MI attacks on two datasets.

**Strengths:**

* This work focuses on defending against black-box MI attacks which have not been addressed in the existing defense.

* As a post-processing defense, SSD can be easily integrated with most pre-trained models for defending against black-box attacks.

* Empirical results demonstrate SSD's success in reducing the attack performance of four state-of-the-art black-box MI attacks.

**Weaknesses:**

* Practical Limitations: Although the idea of post-processing defense is interesting, the proposed method raises concerns about its applicability in real-world scenarios. To modify the model's prediction output, **SSD requires a dataset $D_{valid}$**, which I believe should be real data (either the training dataset or its validation set). This means the user must store raw training data or predictions on the training data to perform predictions, potentially increasing the risk of data leakage.

* SSD's prediction process involves an optimization step for each image, leading to significantly increased computational costs and slower inference times compared to other models.

* The experiments were conducted on low-resolution 64x64 images, limiting the generalizability of the findings high-resolution scenarios.

* [r1] was omitted in the paper while it is also a state-of-the-art black-box attack.

* The paper suffers from some typos and grammatical errors. For examples, lines 41, 107.

[r1] Han, Gyojin, et al. "Reinforcement learning-based black-box model inversion attacks." Proceedings of the IEEE/CVF Conference on Computer Vision and Pattern Recognition. 2023.

**Questions:**

* [r1] was omitted in the paper. I suggest the author should add [r1] in the experiments

* Can the author evaluate the proposed method's performance on high-resolution images? Additionally, please provide more details on how high-resolution MI attacks like MIRROR are adapted to low-resolution scenarios.

* Algorithm 1 references $D_{valid}$ without a clear definition. Please provide an explicit expression for $D_{valid}$. Furthermore, it would be interesting to understand the impact of $D_{valid}$’s size on the final output. Were all training images used as $D_{valid}$?

* For the LOKT experiments, please clarify whether the authors retrained their TACGAN and other surrogate models or used pre-trained models. Note that LOKT requires training target models with GANs and surrogate models,  which in this case SSD needs to be performed on every samples.

*  Please specify the number of identities used for attacks and the quantity of reconstructed images generated per identity.

* I would like to see a comparison of the prediction time between SSD and baseline methods like NoDef or other defenses.

---

> ### Author Response · Authors · 2024-11-22
>
> Thank you for careful review and insightful comments, as well as your appreciation for our method and experiments. We have carefully read all your comments. Here are our replies one by one.
>
>     Please provide an explicit expression for $D_{valid}$.
>
> We employ **Leave-One-Out Cross-Validation (LOOCV)**, where one sample in $D_{train}$ is used to validate for each label. During training the target model, $D_{valid}$ is used to evaluate utility and adjust hyperparameters. After training, the latest $D_{valid}$ is used to estimate the $\pmb{q}^y$ in our defense. Thank you for the inquiry and we will include these details in our paper.
>
>     Furthermore, it would be interesting to understand the impact of $D_{valid}$’s size on the final output. Were all training images used as $D_{valid}$?
>
> Based on the LOOCV setting above, we have $|D_{valid}|=|\mathbb{Y}|=530$ for FaceScrub dataset and $|D_{valid}|=|\mathbb{Y}|=1000$ for CelebA dataset. Your question has sparked our interest, **so we conducted a new experiment by using all training samples as $D_{valid}$**, i.e. $|D_{valid}|=|D_{train}|=27018$ for CelebA. All other experiment settings are the same as the Tables 1~3, where the target model is IR-152 and the dataset is CelebA.
>
> The MIA robustness under Mirror attack:
>
> |          | $\|D_{valid}\|$ | $↓Acc@1$ | $↓Acc@5$ | $↑δ_{eval}$ | $↑δ_{face}$ |
> | -------: | --------------: | -------: | -------: | ----------: | ----------: |
> | Original |            1000 | **1.2%** | **2.9%** |        2527 |    **1.56** |
> |      New |           27018 | **1.2%** |     3.0% |    **2531** |        1.54 |
>
> And the MIA robustness under BREP attack:
>
> |          | $\|D_{valid}\|$ | $↓Acc@1$ | $↓Acc@5$ | $↑δ_{eval}$ | $↑δ_{face}$ |
> | -------: | --------------: | -------: | -------: | ----------: | ----------: |
> | Original |            1000 | **0.4%** | **1.6%** |    **2362** |        1.61 |
> |      New |           27018 | **0.4%** | **1.6%** |        2355 |    **1.62** |
>
> We find that there is almost no difference in MIA robustness between the original and the new.
>
> For model's utility, the evaluation results are
>
> |          | $\|D_{valid}\|$ |    $↑Acc$ | $↓\text{Avg }L_1$ | $↓\text{Max }L_1$ |
> | -------: | --------------: | --------: | ----------------: | ----------------: |
> | Original |            1000 | **90.3%** |          **0.15** |          **0.95** |
> |      New |           27018 |     90.0% |              0.18 |              0.97 |
>
> It can be seen that using $D_{train}$ as $D_{valid}$ leads to a decrease in model utility. We think the reason is: the estimating of $\pmb{q}^y$ by $D_{train}$ is not as accurate as the one by $D_{valid}$. However, the decrease is very minor and acceptable. **If the model owner has not prepared validation set, we suggest they could use training set as validation set.**
>
>     SSD requires $D_{valid}$, which I believe should be real data (either the training dataset or its validation set). This means the user must store raw training data or predictions on the training data to perform predictions, potentially increasing the risk of data leakage.
>
> Under our careful design, **model owners need not to store the complete $D_{valid}$, nor the raw predictions of private samples**. In fact, they only need to store the **average prediction** for each label, i.e.
> $${\tilde{\pmb{q}}}^{y}\coloneqq\underset{(\pmb{x},y)\in D_{valid}}{\text{mean}}{\pmb{f}(\pmb{x})}$$
> for each $y\in\mathbb{Y}$. We believe that the averaging operation can greatly mitigate the risk of data leakage [1].
>
>     SSD's prediction process involves an optimization step for each image, leading to significantly increased computational costs and slower inference times compared to other models.
>
>     I would like to see a comparison of the prediction time between SSD and baseline methods like NoDef or other defenses.
>
> As we stated in [Common Concerns on Computational Cost](https://openreview.net/forum?id=p0DjhjPXl3&noteId=aptS4M72M9), we have completely addressed this problem. **Modifying 512 predictions for 100 times only needs 0.5 seconds with our careful optimization.**
>
> ---
>
> The experiments about RLB-MI and high-resolution are ongoing. The results will be available before the end of the discussion.
>
> ---
>
> [1] Dwork, Cynthia. “Differential Privacy.” International Colloquium on Automata, Languages and Programming (2006).

---

> > ### Comment · Reviewer_VqZg · 2024-11-22
> >
> > I thank the authors for the additional results and details.
> > > We employ Leave-One-Out Cross-Validation (LOOCV), where one sample in $D_{train}$  is used to validate for each label.
> >
> > > Based on the LOOCV setting above, we have |$D_{valid}$| = |Y| = 530 for FaceScrub and |$D_{valid}$| = |Y| = 1000 for CelebA
> >
> > > In fact, they only need to store the average prediction for each label
> >
> > Your response is a bit unclear to me. As I understand, $D_{valid}$ is used to compute $\tilde{q}^y$ via Equation (2). $D^y$ is a subset of $D_{valid}$ containing only samples labeled as class $y$. $\tilde{q}^y$ represents the average prediction of all samples within $D^y$.
> >
> > If |$D_{valid}$| = |Y| = 530 for FaceScrub and |$D_{valid}$| = |Y| = 1000 for CelebA, does this mean there's only **one sample per class** in |$D_{valid}$|? If so, is |$D^y$| = 1 in Equation (2)? And you are not storing the **average prediction** for each label but **the prediction for each sample**?

---

> > > ### Author Response · Authors · 2024-11-22
> > >
> > > In our algorithm, we propose to use average prediction for accurately estimating $\pmb{q}^y$ and protecting privacy. In our experiment, we use one validation sample per label because some labels have few samples (e.g., some labels in FaceScrub only have 4 training samples). The latter is a special case of the former.
> > >
> > > We continue to answer your other questions below.
> > >
> > >     For the LOKT experiments, please clarify whether the authors retrained their TACGAN and other surrogate models or used pre-trained models. Note that LOKT requires training target models with GANs and surrogate models, which in this case SSD needs to be performed on every samples.
> > >
> > > Yes. We know that LOKT's GAN training is dependent on the target model. Whenever the target model or defense changed, we retrain TACGAN and three surrogate models.
> > >
> > >     Please specify the number of identities used for attacks and the quantity of reconstructed images generated per identity.
> > >
> > > The public dataset is FFHQ, and all identities in CelebA and FaceScrub are under attack. For each identity, we reconstruct 5 images. We will note these in the paper.
> > >
> > > Experiments are underway. If you have any other questions, please feel free to ask us.

---

> ### Author Response · Authors · 2024-11-25
> **High resolution and RLB results**
>
> We are pleased to reply to you with the results for high resolution and RLB, once again proving the superiority of our defense. We now address your remaining concerns.
>
>     please provide more details on how high-resolution MI attacks like MIRROR are adapted to low-resolution scenarios.
>
> To adapt to low resolution, we use StyleGAN2 trained on FFHQ with a resolution of 256×256 [1]. The generated images are center-cropped to 176×176, resized to 64×64, and input into the target model. All other attack settings are consistent with the original code.
>
>     Can the author evaluate the proposed method's performance on high-resolution images?
>
> Yes! We chose Mirror as the attacker. We use StyleGAN2 trained on FFHQ with a resolution of 1024×1024 [1]. The generated images are center-cropped to 800×800, resized to 224×224, and input into the target model. The target model is ResNet-152 and the evaluation model is Inception-v3. The first 10 classes of FaceScrub are attacked and each class reconstructed 5 images. The attack results are:
>
> | Defense  | $↓Acc@1$ | $↓Acc@5$ | $↑δ_{eval}$ | $↑δ_{face}$ |
> | :------: | -------: | -------: | ----------: | ----------: |
> |    No    |      70% |      94% |         195 |        0.84 |
> |          |          |          |             |             |
> |   MID    |      62% |      90% |         183 |        0.76 |
> |   BiDO   |      66% |      86% |         194 |        0.90 |
> |    LS    |      48% |      82% |         202 |        0.87 |
> |    TL    |      58% |      92% |         191 |        0.80 |
> | **Ours** |  **42%** |  **66%** |     **211** |    **1.13** |
>
> It can be seen that **our defense has the best MIA robustness in high-resolution scenarios.**
>
> The target models' utility and defenses' hyperparameters are:
>
> | Defense | $↑$Test Acc | $↓\text{avg }L_{1}$ | $↓\text{max }L_{1}$ |          Hyperparameters           |
> | :-----: | ----------: | ------------------: | ------------------: | :--------------------------------: |
> |   No    |       98.5% |                   0 |                   0 |                 -                  |
> |         |             |                     |                     |                                    |
> |   MID   |       96.7% |                0.30 |            **1.97** |           $\beta=0.005$            |
> |  BiDO   |       96.3% |                0.09 |                1.99 | $\lambda_{x}=0.15,\lambda_{y}=1.5$ |
> |   LS    |       96.5% |                0.11 |                1.99 |           $\alpha=-0.01$            |
> |   TL    |       96.7% |                0.19 |                1.99 |          First 70% layers          |
> |  Ours   |   **96.9%** |            **0.07** |                1.98 |        $T=1,\varepsilon=20$        |
>
> Due to the deadline, we have no time to adjust our hyperparameters to achieve the best $\text{max }L_{1}$. However, other metrics are sufficient to prove that **we have the best model's utility in high-resolution scenarios.**
>
>     [r1] was omitted in the paper. I suggest the author should add [r1] in the experiments
>
> Following your suggestion, we supplement a new experiment on RLB. All attack settings are consistent with the original code [2]. The target model is IR-152, the evaluation model is Inception-v3, and the GAN is trained on FFHQ. The first 10 classes of CelebA are attacked and each class reconstructed 5 images. The attack results are:
>
> | Defense  | $↓Acc@1$ | $↓Acc@5$ | $↑δ_{eval}$ | $↑δ_{face}$ |
> | :------: | -------: | -------: | ----------: | ----------: |
> |    No    |      32% |      64% |        2006 |        0.77 |
> |          |          |          |             |             |
> |   MID    |      30% |      48% |        2088 |        0.84 |
> |   BiDO   |      16% |      28% |        2254 |        0.94 |
> |    LS    |      12% |      34% |        2204 |        0.85 |
> |    TL    |      22% |      34% |        2107 |        0.82 |
> | **Ours** |   **8%** |  **12%** |    **2480** |    **1.26** |
>
> It can be seen that **our defense has the best MIA robustness against RLB.** The target models' utility and defenses' hyperparameters are consistent with the Table 3~4 in our paper, which shows that **we have the best model's utility, too.**
>
> ---
>
> The above two new experiments show that both in high-resolution and RLB-attack scenarios, our defense remains the best. We will add them to our appendix. If you have any other questions, please feel free to ask us.
>
> ---
>
> [1] https://github.com/NVlabs/stylegan2
>
> [2] https://github.com/HanGyojin/RLB-MI

---

> > ### Comment · Reviewer_VqZg · 2024-11-27
> >
> > Thank you for your response.
> >
> > > In our algorithm, we propose to use average prediction for accurately estimating $q^y$ and protecting privacy. In our experiment, we use one validation sample per label because some labels have few samples (e.g., some labels in FaceScrub only have 4 training samples).
> >
> > In my view, it is inconsistent to propose computing the average prediction $q^y$ while reporting results based on $q^y$ from **a specific sample**. I suggest aligning the experimental setting with the proposed method.
> >
> > > |LS	|96.5%|	0.11|	1.99	| $\alpha$ = 0.01|
> >
> > To clarify, are the reported results in this rebuttal for LS based on positive LS (α = 0.01) or negative LS?

---

> ### Author Response · Authors · 2024-11-27
>
> Thank you for your advice! We will maintain consistency in the revised paper.
>
>     To clarify, are the reported results in this rebuttal for LS based on positive LS (α = 0.01) or negative LS?
>
> We set $\alpha=-0.01$ for LS. Thank you for pointing out this typo and we have corrected it in the rebuttal.

---

> ### Author Response · Authors · 2024-12-01
>
> We appreciate your feedback and contribution to improve our work! If all your questions have been addressed (validation set, time overhead, high resolution, and RLB attack), we kindly ask you to consider raising your rating. If you still have any doubts or reservations about our work, we are more than willing to engage in further discussion with you.

---

> ### Author Response · Authors · 2024-12-03
>
> The discussion will end in 10 hours. If you still have any doubts or reservations (including but not limited to RLB attacks, high resolution, time consumption, validation sets), please leave a message before it ends. We are more than happy to respond promptly.

---

> > ### Comment · Reviewer_VqZg · 2024-12-03
> >
> > I appreciate the authors' efforts to address my questions and update the paper. I have increased the score accordingly.
> >
> > However, my primary concern regarding privacy leakage persists. **SSD requires the SoftMax output on private samples (which is no longer an average prediction in the updated version)**, potentially increasing the risk of data leakage. It's important to consider that this risk can originate from both **internal adversaries** (e.g., system administrators) and **external adversaries** (e.g., API users).
> >
> > Typically, in white-box attack and defense scenarios, both types of adversaries possess equivalent access to the system. In contrast, for black-box defenses, internal adversaries can leverage white-box attacks, while external adversaries are limited to black-box or label-only attacks.
> >
> > For this proposed method, a crucial question arises: **Could internal adversaries exploit SoftMax output on private samples $q^y$ to launch white-box attacks, potentially increasing data leakage?** This warrants further investigation.

---

> > > ### Author Response · Authors · 2024-12-03
> > >
> > > Thank you for appreciating our efforts and recognizing our work!
> > >
> > > The only reason for using one prediction rather than average is that: our validation set is not large enough. If it has more samples per class, the average prediction will be used. In Appendix E of our revision, we discuss the case of using the training set as validation  and using the average prediction. We have shown that it also has a good defense and utility. In our final version, we will enlarge our validation set and use average predictions.
> > >
> > > The external adversaries and internal adversaries that you raised are very interesting new concepts. Previous research only considered external adversaries, without taking into account internal traitors. The latter may be worth researching in the future. In our work, whether one prediction or average prediction, there is only one for each class. This information is too little for internal traitors.
> > >
> > > Finally, we thank all reviewers for their appreciation, support, encouragement and contribution! All your contributions will be reflected in our final paper.

---

### Author Response · Authors · 2024-11-21
**Common Concerns on Computational Cost**

We sincerely appreciate all the reviewers for dedicating their valuable time and effort to review our paper. Their insightful comments and suggestions are conducive to improving our paper.

Encouragingly, the reviewers praise the significance of black-box defense (R#VqZg), the generalizability of our post-processing (R#VqZg, R#KsSS and R#J1L4), the insight of using conditional mutual information (R#Ui99, R#KsSS and R#J1L4), the rigor of our mathematical proof (R#Ui99 and R#J1L4), and the effectiveness shown by our experiments (all reviewers).

Below, we uniformly address the concerns shared by more than one reviewers.

---

## Common Concerns on Computational Cost (R#VqZg and R#J1L4)

We formulate the minimization of CMI as a convex optimization problem. Our post-processing module needs to solve it for each prediction, which raises concerns about computational overhead. We indeed encountered this problem before. At that time we turned to CVXPY [1], a famous convex optimizer, but its efficiency and accuracy are not satisfactory. Fortunately, this concern has been solved through our efforts:

**Theoretically, we propose to solve the convex optimization problem via water-filling.** After a series of mathematical derivations (presented in appendix), we find the explicit form of the optimal solution. Our water-filling algorithm can reach the solution with the time complexity of $O(|\mathbb{Y}|\log|\mathbb{Y}|)$, where $|\mathbb{Y}|$ is the number of labels.

**Practically, we implement our water-filling on GPUs.** We manage to replace the branching and looping operations with sequential vectorization operations, and then implement our algorithm on PyTorch Tensors (provided in supplementary material). Our GPU-based-water-filling can be regarded as a post-processing layer of neural networks, whose computational cost is not greater than the other layers.

Following the reviewers' suggestions, **we add a quantitative experiment.** The models, datasets, and defense settings are the same as the Table 3 in our paper. We take a batch with 512 samples and let the model infer 100 times on it. We record the time cost by `torch.profiler`, an official tool provided by PyTorch. We exclude the time for I/O (i.e. the time from disk to memory, and from CPU to GPU), and only include the time for forward propagation on GPU. Our experiment is conducted on one NVIDIA GeForce RTX 3090. Here is the result:

|                           | IR-152 & CelebA | IR-152 & FaceScrub | VGG-16 & FaceScrub |
| :-----------------------: | :-------------: | :----------------: | :----------------: |
|   Time without defense    |     18.63 s      |       17.70 s       |       5.65 s        |
|   Time with our defense   |     19.22 s      |       18.16 s       |       6.07 s        |
| Percent of increased time |      3.1%       |        2.5%        |        7.4%        |

It can be seen that we only increased the time by 2.5% to 7.4%. The higher percent on VGG is due to the simpler model structure. In absolute terms, **modifying 512 predictions for 100 times only needs 0.5 seconds. If we take the I/O time into account, the percents are small enough to be ignored.** We will add this experiment to the appendix.

---

[1] Steven Diamond and Stephen Boyd. CVXPY: A Python-embedded modeling language for convex optimization. Journal of Machine Learning Research, pp. 1–5, 2016.

---

> ### Comment · Reviewer_J1L4 · 2024-11-26
>
> Thank you for providing a detailed response that addresses the concerns regarding computational cost. I appreciate the theoretical and empirical efforts you have undertaken to address the issue. Including them in the appendix will help clarify this point for potential readers.

---

### Author Response · Authors · 2024-11-21
**Common Concerns on Adaptive Attacks**

We sincerely appreciate all the reviewers for dedicating their valuable time and effort to review our paper. Their insightful comments and suggestions are conducive to improving our paper.

Encouragingly, the reviewers praise the significance of black-box defense (R#VqZg), the generalizability of our post-processing (R#VqZg, R#KsSS and R#J1L4), the insight of using conditional mutual information (R#Ui99, R#KsSS and R#J1L4), the rigor of our mathematical proof (R#Ui99 and R#J1L4), and the effectiveness shown by our experiments (all reviewers).

Below, we uniformly address the concerns shared by more than one reviewers.

---

## Common Concerns on Adaptive Attacks (R#KsSS and R#J1L4)

Reviewers are concerned about adaptive attacks, where attackers know our defense and specifically design their attack strategies. We appreciate them for raising such an interesting question, which has stimulated our deeper thinking.

Firstly, we believe that launching adaptive attacks in black-box scenarios is unrealistic, because attackers don't know the target model, and naturally don't know its defense strategy. If they were to guess the defense strategy based on the model's behavior, they would need to consume a large number of queries.

Step back and consider, **if attackers know our defense, the best attack strategy we think is:**
1. Query the same $\pmb{x}$ repeatedly and count the frequency of different outputs.
2. Estimate our sampling probability $\mathbb{P}(y|\pmb{x})$ by the frequency.
3. Infer our true prediction $\mathbb{P}(\hat{y}|\pmb{x})$ based on $\mathbb{P}(y|\pmb{x})$ and temperature $T$ (assuming they know).

If an online server detects such pattern of queries, it can block them. Step back and consider again, **we now propose a memory-free and low-cost improvement against adaptive attacks:**

Design a hash function $h:\mathbb{X \rightarrow N}$, where $\mathbb{X}$ is the input space and $\mathbb{N}$ is the set of integers. When users/attackers query $\pmb{x}$, we take $h(\pmb{x})$ as a random seed for sampling on the probability $\mathbb{P}(y|\pmb{x})$. This ensures *same-input-same-output*, thus preventing the adaptive attackers. However, attackers can add subtle perturbations to $\pmb{x}$, therefore our $h$ needs to be robust. For example, it can be
$$ h(\pmb{x}) \coloneqq \sum_{i = 1}^{m}\lfloor k \cdot z_{i}(\pmb{x}) \rfloor $$
where $\pmb{z}(\pmb{x}){\in}\mathbb{R}^{m}$ is the penultimate layer feature in target model, and $k$ is the sensitivity coefficient. Note that $\pmb{z}(\pmb{x})$ are commonly used to assess the similarity between different images, i.e., **the closer the two $\pmb{z}(\pmb{x})$ are, the more similar the two $\pmb{x}$ look.** We map $\pmb{z}(\pmb{x})$ to integers by rounding down and summing up. The larger $k$ is, the more numerically sensitive $h$ is, and the more random our defense is.

How to evaluate and improve $h$ is a new topic, and we believe it can be an independent research work. Thanks again to the reviewers for pointing out adaptive attacks, and we will supplement the discussion above in our paper.

---

> ### Comment · Reviewer_J1L4 · 2024-11-26
>
> Thank you for providing us a detailed comment regarding the concerns about adaptive attacks. I appreciate your explanation and the effort to propose a memory-free, low-cost improvement to enhance the robustness of your defense mechanism.
>
> You make a valid point about the practical challenges attackers encounter in black-box scenarios. Indeed, launching adaptive attacks in such settings requires significant resources, and your argument about the realistic limitations attackers encounter strengthens the case for the feasibility of your defense. Furthermore, the introduction of the hash function h(x) to randomize the sampling process and ensure "same-input-same-output" is an intriguing idea. The connection to the similarity between features through z(x) is particularly compelling, as it aligns well with the intuitive understanding of feature-space robustness. I look forward to seeing these ideas expanded in the final version of the paper.

---

> > ### Author Response · Authors · 2024-11-27
> >
> > Thank you for your appreciation on our solution to adaptive attacks! We will expand these ideas in our paper.

---

### Meta-Review · Area_Chair_QxMX · 2024-12-22

**Metareview:**

This paper presents a new defense against model inversion attacks. The main idea is to post-process the model's output by minimizing the conditional mutual information between the predicted probability vector and the input. The authors show experimentally that their defense can effectively mitigate state-of-the-art black-box attacks. While reviewers raised several weaknesses (see below), most of them are addressed by the rebuttal, and AC is happy to recommend acceptance conditioned on these changes being included in the camera ready version.

**Additional Comments On Reviewer Discussion:**

Reviewers raised several weaknesses, including:
1. Increased computation cost during inference.
2. Proposed defense may be vulnerable to adaptive attacks and/or attacks executed by an insider.
3. Lack of evaluation on deep model and high-dimensional data.

In the rebuttal, the authors addressed Weakness 1 with timing experiments that showed the overhead is small, provided an analysis of adaptive attacks to address Weakness 2, and included additional experiments to address Weakness 3.

---

### Decision · Program_Chairs · 2025-01-22

Accept (Poster)